# Factors influencing the temporal variability of atmospheric methane emissions from Upper Silesia coal mines: A case study from CoMet mission.

Justyna Swolkień[1], Andreas Fix[2] and Michał Gałkowski[3,4]

[1]Faculty of Civil Engineering and Resource Management, AGH University of Science and Technology, Krakow, Poland
[2]Deutsches Zentrum für Luft- und Raumfahrt (DLR), Institut für Physik der Atmosphäre, Oberpfaffenhofen, Germany
[3]Max Planck Institute for Biogeochemistry, Jena, Germany
[4]Faculty of Physics and Applied Computer Science, AGH University of Science and Technology, Krakow, Poland

*Correspondence to*: Justyna Swolkień (swolkien@agh.edu.pl)

**Abstract:**

Methane is a powerful greenhouse gas responsible for around 20% of radiative forcing (relative to the pre-industrial era) caused by all long-lived greenhouse gases (WMO 2021). About 60% of the global emissions are from anthropogenic sources and coal mining is one of the largest contributors. Emissions are either estimated by bottom-up approaches (based on inventories) or top-down approaches (based on atmospheric measurements). Combining those with an accurate error estimation allows to better characterise model errors e.g. caused by transport mechanisms.

Here we provide a detailed description of factors influencing the coal-mine methane emissions variability. We use high-frequency (up to hourly) temporal data from seven coal mines in the Upper Silesian Coal Basin during the CoMet 1.0 (Carbon dioxide and Methane) mission from May 14 to June 13, 2018. The knowledge of these factors for the individual ventilation shaft is essential for linking the observations achieved during the CoMet 1.0 with models, as most publicly available data in the bottom-up worldwide inventories provide annual emissions only.

The methane concentrations in examined shafts ranged from 0.10 to 0.55 ± 0.1% during the study period. Due to the changing scope of mining works performed underground, they were subjected to a significant variation on a day-to-day basis. The yearly methane average emission rate calculated based on one month set of temporal data of the analysed subset of mines was of the order of 142.68 kt yr$^{-1}$ (σ = 18.63 kt yr$^{-1}$), an estimate lower by 27 % than the officially published WUG (State Mining Authority) data and 36 % than reported to E-PRTR (European Pollutant Release and Transfer Register). We also found that emissions from individual coal mine facilities were over- and underestimated between 4 % to 60 %, compared to E-PRTR, when short-term records were analysed. We show that the observed discrepancies between annual emissions based on temporal data and public inventories result from 1) the incorrect assumption that the methane emissions are time-invariant 2) from the methodology of measurements, and lastly, 3) from the frequency and timing of measurements.

From the emission monitoring perspective, we recommend using a standardised emission measurement system for all coal mines, similar to the SMP-NT/A (Methane Fire Teletransmission Monitoring System). Legal safety requirements require all coal mines to implement this system. After an adaptation, the system could allow for gas flow quantification, necessary for accurate and precise estimations of methane emissions at a high temporal resolution. Using this system will also reduce the emission uncertainty due to factors like frequency and timing of measurements.

In addition, it would be beneficial to identify separately the emissions from individual ventilation shafts and methane drainage stations. That would bridge the gap between bottom-up and top-down approaches for coal mine emissions. The intermittent releases of unutilised methane from the drainage stations are currently not considered when constructing regional methane budgets.

## 1. Introduction

Methane is a greenhouse gas emitted from a wide variety of highly dispersed sources that overlap geographically. Methane has a lower atmospheric concentration relative to $CO_2$, but its global warming potential (GWP) is 28 times higher (on a 100-year horizon) than that of $CO_2$ (Myhre et al., 2013; IPCC, 2014). More recent studies suggest the GWP be even 34 on a 100-year horizon and 86 on a 20-year horizon (Myhre et al., 2013; Etminan et al., 2016). Although its global emission accounts for about 4 % of the anthropogenic $CO_2$ emission in mass flow units it is

nevertheless responsible for ~ 20 % of the additional radiative forcing accumulated in the lower atmosphere since 1750 (Saunois et al., 2020). Our ability to accurately predict future climate change is still challenged by our failure to identify precisely methane emissions sources. Therefore, reducing the level of uncertainty in the estimated amount of released methane remains of utmost importance across a wide array of natural and anthropogenic emission sources, including coal mines.

To that end, attempts at improving the emissions estimates are generally following two methodologies: 1) where emissions are obtained and subsequently aggregated from available activity data (the so-called bottom-up approach) or process-based models 2) where the results of direct atmospheric observations (from the ground, aircraft, and satellites) are used together in tandem with atmospheric transport models of varying complexity to obtain those emission estimates - the so-called top-down approach (e.g. Saunois et al., 2020, 2016).

Bottom-up approaches are used to create publicly available databases such as UNFCCC Greenhouse Gas Inventory Data (*United Nations Framework Convention on Climate Change*), EDGAR (*Emissions Database for Global Atmospheric Research*), or the UNFCCC Scarpelli database (UNFCCC database, Scarpelli et al., 2020; Höglund-Isaksson 2012; Saunois et al., 2020). These inventories are based on different assumptions and use different data sets in calculations but follow the same IPCC Guidelines. For instance, UNFCCC provides $CH_4$ emissions from

underground coal mines, which are reported in accordance with the guidelines for Annex I and non-Annex I countries. In Poland, these values are provided by the State Mining Authority inventory (Wyższy Urząd Górniczy, WUG, 2019). On the other hand, the EDGAR database compiles consistent anthropogenic global emissions and trends based on international statistics and technology-based emission factors for use in atmospheric models and policy evaluation.

Additionally, EDGAR provides global emission estimates, disaggregated at the source-sector level (Crippa et al.,

2021; Janssens-Maenhout et al., 2019). As a result, the databases mentioned above require a vast amount of country-specific information, and if it is not available, coefficients suggested by IPCC are used instead (Höglund-Isaksson 2012). Valuable inventories are also available on the regional scale. In 2006, in response to the Aarhuus Convention, the European Union created the E-PRTR database (European Pollutant and Transfer Register). It contains emission data reported annually by individual facilities from 27 European Member States whose emissions exceed the threshold

of 1 kt $CH_4$ per year.

Top-down approaches involve methods based on state-of-the-art measurement technologies using satellites, ground-based measuring devices, and aircraft (Gurney et al., 2002; Houweling et al., 2017; Fix et al., 2018a; Bergamaschi et al., 2018; Saunois et al., 2020). Currently, the existing possibilities for monitoring the concentration of greenhouse gases can be divided into in-situ methods, where the local air composition is characterized by instruments located in the measurement location, on the ground or aboard an aircraft (for the latter see e.g. Kostinek et al., 2020, Fiehn et al., 2020, Galkowski et al., 2021), or remote sensing methods, where the composition across a larger air volume is characterized by measurements of electromagnetic radiation passing through the targeted airmass. Regardless of their placing (ground based, airborne or spaceborne), most of the remote sensing instruments rely on sunlight as a source of radiation (e.g. Bovensmann, 2019, Luther et al. 2019), but measurements of greenhouse gases using active sensors (lidars) have also been reported recently (Amediek et al., 2017, Fix et al., 2020).

To implement an integrated methane emission monitoring system, top-down and bottom-up approaches should support and complement each other. However, one of the most critical issues is obtaining temporally-resolved emissions from individual point sources (Swolkień, 2020). Bottom-up inventories cover either annual (UNFCCC, E-PRTR, WUG) or monthly (EDGAR) aggregates of released $CH_4$ (UNFCCC, E-PRTR), allowing to verify measurement data based on e.g. in-situ aircraft measurements with a level of consistency but only on a regional scale. Fiehn et al., 2020 and Kostinek et al., 2020 compared the entire USCB using a mass balance and model-based approach during CoMet 1.0. In the first case, the authors showed that $CH_4$ emissions estimates from two flights were in the lower range of the six presented emission inventories (Fiehn et al., 2020). In the second case, the obtained emission rates coincided ($\pm 2$ %) with annual-average inventorial data from E-PRTR 2017, but they were distinctly lower (-37 % / -40 %) than values reported in EDGAR v4.3.2 (Kostinek et al., 2020). A significant problem in comparing the results of top-down measurements arises when attempts are made to identify emissions from individual sources, such as ventilation shafts. Using annual data for this purpose may lead to overestimating or underestimating emissions from individual sources. We will show that in the course of this paper.

Temporally resolved data at higher frequencies are unavailable in any of the existing databases, despite being necessary to provide estimates more accurate than standard inventory data (Swolkień, 2020). In addition, they could also improve predictions of high-resolution atmospheric transport modelling. Therefore, all efforts to monitor the temporal evolution of individual sectors, not just yearly averages, need more support and will significantly advance the National Reporting to the UNFCCC. Moreover, temporal variation of emissions on various time scales can provide additional constraints to distinguish methane emissions from different sectors.

As methane emissions into the atmosphere are a matter of great concern, a multitude of available methods are employed across the globe to conduct large-scale research geared towards introducing an integrated system for monitoring greenhouse gas emissions (Fix et al. 2018a). An example of a campaign aimed at improving the methodology for measuring the efficiency of gas streams ($CO_2$, $CH_4$) at a local and regional scale (Fix A. et al., 2015; Fix et al., 2018b; Fix et al., 2020) was CoMet 1.0 (Carbon dioxide and Methane) (Fix et al. 2020; Luther et al. 2019; Galkowski et al. 2021). The campaign combined active (lidar) and passive (spectrometer) remote sensors with in-situ instruments installed onboard a German research aircraft HALO (High Altitude and Long Range Research Aircraft, Gulfstream G550) and two smaller airplanes (Cessna 208 Grand Caravan, Cessna 207). Additional support was

provided by stationary and mobile ground based measurements, as well as drones (Luther et al., 2019; Fix et al., 2018a; Bovensmann et al., 2019; Andersen et al., 2021; Andersen et al., 2022).

The mission's goal was to comprehensively measure the distribution of greenhouse gases and, in conjunction with modelling activities, to investigate and improve methodologies to estimate local and regional fluxes of greenhouse gases from anthropogenic sources such as coal mining. The CoMet team chose the USCB as an interesting test case for these investigations.

        The paper's primary purpose is to explain the nature of methane emission from Polish underground coal mines
localised in the USCB and its temporal variability in order to link the mining and atmospheric science perspectives. Furthermore, we describe the factors that simultaneously exert significant influence on the coal-mine methane emissions variability, emphasising the need for obtaining temporally-resolved data to improve the quality of the available bottom-up inventories. The awareness of factors affecting methane concentrations variability in individual ventilation shafts (and thus the amount of methane emitted into the atmosphere) is essential for the accuracy and
precision of independent coal-mine emissions estimations driven by atmospheric observations (like those collected during CoMet 1.0 campaign). Conclusions drawn from this study can be used to improve atmospheric models, as well as to prepare for future exercises, targeting other types of point sources, or their regional clusters.

        In addition, based on temporal methane emissions analysis, we indicate the possibility of using the methane fire monitoring system, after prior adaptation, as a standardised method for methane emissions quantification in all coal
mines.

        The first part of the article (chapters 2 and 3) explains the adoption of USCB as a research area during the CoMet 1.0 mission. The second (chapter 4) describes in detail the methodology of methane concentration measurements in selected coal mines. Finally, the third part (chapters 5 and 6) describes and explains different factors affecting the variability of concentrations and fluxes from ventilation shafts. Also, in this part we compare the temporal data with
WUG and E-PRTR inventories and set out guidelines for using a standardised methane monitoring system for all coal mines.

## 2     Description of the methane emissions from the USCB area and their evaluation

        Methane emitted from underground coal-seams mines is generally referred to as CMM (Coal Mine Methane), or
total methane bearing capacity or total methane emission. In the studies focused on safety aspects of methane in the mines (Swolkień 2015; Boger et al., 2014; Karacan et al., 2011), CMM is often used next to AMM (Abandoned Mine Methane), which denotes same type of emissions in abandoned mines.

        It is important to note that in some cases not all of CMM is vented directly into the atmosphere via ventilation shafts. If the methane drainage system is installed, a part of methane is captured by it and transported through a special
distribution system to utility installations (e.g. local gas power plants). The methane that is not captured is a well-know safety risk and needs to be vented (VAM – ventilation air methane) to the atmosphere via ventilation shafts.

        Methane emissions from coal-mining are closely related to coal production. Poland is the tenth-largest coal producer globally, with domestic extraction reaching 63.4 million tonnes (IEA 2019) in 2018. At the same time, Polish emissions of $CH_4$ from underground mining are ranked seventh in the world, with 2 % of the total methane emitted

from this source category in 2018 (UNFCCC database). According to the 2020 National Inventory Report (NIR), the total $CH_4$ emission in Poland in 2018 amounted to 1950 kt, representing a decrease of 35.6 % from the base year (1988). Additionally, over 30 % of the country's total $CH_4$ emissions this year came from the mining sector (NIR 2020). Poland is the most significant contributor of coal-mining methane in Europe, responsible for 604 kt released into the atmosphere according to UNFCCC, corresponding to 41 % of the total European emissions from this sector. Other significant contributions to the total $CH_4$ emission came from Ukraine (35 %), Romania (14 %), Germany (4 %) and Czechia (3 %), the remainder being small contributions from other countries. It is worth mentioning that these statistics do not include the Russian Federation, which emitted 1462 kt in 2018, as separate data from the European part of the country is not available from UNFCCC (https://di.unfccc.int/detailed_data_by_party)

According to Polish State Mining Authority (WUG) in Poland, atmospheric methane emissions (including both ventilated and non-utilised methane) from underground mining reached 511 kt $CH_4$ in 2018 (Table 1) (WUG, 2019). This number is not directly comparable to emissions reported to UNFCCC, as the latter includes atmospheric methane emissions (so-called mining activities), post-mining activities and abandoned coal mines all together. The last two quantities are determined based on the UNFCCC emissions factors.

Most mining-related methane emissions in Poland originate from the Upper Silesia Coal Basin (USCB), the country's largest industrial district, which is heavily dependent on the mining industry. According to Poland's 2019 balance of mineral resources and underground water, methane release and capture in hard coal seams has been appropriately recorded only in the Upper Silesian Coal Basin (PIG-PIB, 2020). No detailed examination is available of methane resources in the collieries of the other two major coal excavation regions is available (Lower Silesian Coal Basin and the Lublin Coal Basin). A reconnaissance study by Szlazak et al. (2014) described measurements of methane content carried in these areas based on laboratory tests of drill cuttings from boreholes drilled from the surface. These studies revealed that methane content levels in the latter mining areas are considerably lower, which makes it difficult to assess their economic significance.

As of 2018, 29 active coal mines were operational in USCB, covering an area of 5600 $km^2$ (WUG, 2019), six of which were in the state of liquidation, meaning a slow reduction of activity towards shutdown, usually lasting several years. Figure 1 presents the map of the mining areas of the USCB with the indication of the mines with verified methane releases (WUG, 2019) and ventilation shafts targeted during the CoMet 1.0 mission. It is important to note that due to restructuring efforts undertaken in the past 30 years, coal mines can nowadays operate either as an individual facility (e.g. Pniówek coal mine) or within an enterprise ('combined entity') that usually consists of two or three individual facilities (e.g. Knurów-Szczygłowice coal mine). In the latter case, the individual facilities are referred to as fronts (Knurów front and Szczygłowice front). It is challenging to track emissions from individual fronts over the years because emission reporting is only required at the enterprise level, i.e. in case of combined entity the annual emissions are usually reported as sum from individual fronts. For clarity, when 'coal mine' term is used throughout this study, an individual facility is meant unless otherwise stated.

The geological structure of the deposit located in the USCB region favours gas migration. Methane residing in the coal bed is a product of carbonisation of organic substances. The specific methane emission (SME) is the potential to release methane per 1 tonne of extracted coal. It can be calculated by dividing the total methane bearing capacity in a

given year by the annual production. For Polish coal deposits, it reached 14.40 m$^3$ t$^{-1}$ (see Table 1) in 2018, and in the years 2013-2018, it increased from 11.10 to 14.40 m$^3$ t$^{-1}$. The value of SME factor, together with coal output, is used as activity data for calculating atmospheric methane emissions in inventories such as UNFCCC. Apart from Poland, mines characterised by high SME are abundant in Russia, Ukraine, China, the United States of America, and the Czech Republic, among others (IPCC, 2014).

The USCB inventory estimates available in E-PRTR (European Pollutant Release and Transfer Register) for 2018 show that depending on the individual mining facilities or fronts, CH$_4$ emissions amounted to 2.00 kt and 78.40 kt (E-PRTR, 2018). On the other hand, WUG stated that the CH$_4$ emissions for individual mining facilities and combined entities ranged between 0.14 kt and 66.14 kt (WUG, 2019) in that year. It is worth noting that the two emission inventories differ slightly in the methodology of compiling the results. In the WUG inventory, the atmospheric methane emissions are calculated based on individual facilities and combined entities' ventilation air methane and the amount of non-utilised methane for the whole mining sector. On the other hand, the E-PRTR database consists of the complete methane emission data (ventilation air methane plus the amount of non-utilised methane) registered for individual facilities and fronts.

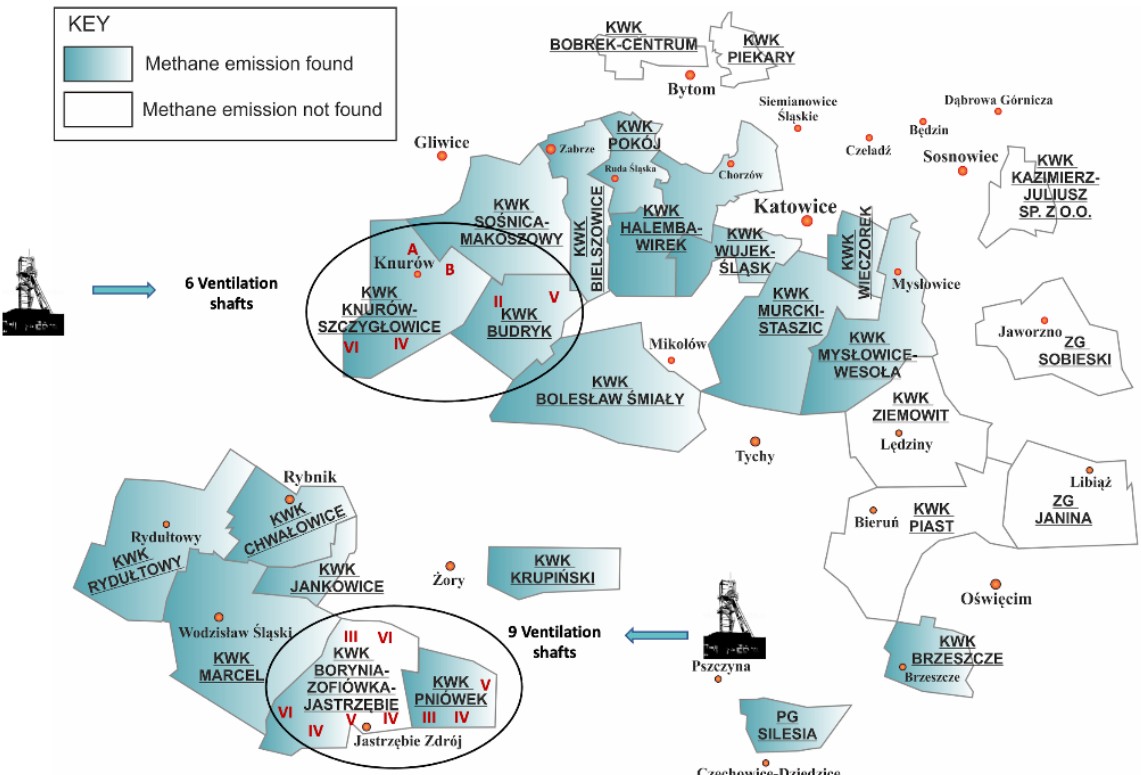

**Figure 1.** Map of mining areas of the Upper Silesia Coal Basin with the indication of the mines with verified methane release (demonstrative drawing). Also, the underground coal mines in the case study and their ventilation shafts are marked, as of 2018.

**Table 1.** Methane emission from Polish underground coal mines from 2013 to 2018 (WUG, 2019)

|  | 2013 | 2014 | 2015 | 2016 | 2017 | 2018 |
|---|---|---|---|---|---|---|
| Total methane bearing capacity, kt yr$^{-1}$ | 615.40 | 638.92 | 668.96 | 669.53 | 680.07 | 656.84 |
| Ventilation air methane, kt yr$^{-1}$ | 417.08 | 408.76 | 425.90 | 424. 5 | 438.45 | 429.55 |

| | | | | | | |
|---|---|---|---|---|---|---|
| Drained methane, kt yr$^{-1}$ | 198.32 | 230.16 | 243.06 | 245.29 | 241.63 | 227.29 |
| Amount of utilized methane, kt yr$^{-1}$ | 134,.8 | 151.57 | 141.32 | 139.82 | 152.00 | 145.62 |
| Atmospheric methane emission, kt yr$^{-1}$ | 480.82 | 487.34 | 527.64 | 529.72 | 528.07 | 511.22 |
| Drainage efficiency | **32%** | **36%** | **36%** | **37%** | **36%** | **35%** |
| Percentage of methane released to the atmosphere[a] | **78%** | **76%** | **79%** | **79%** | **78%** | **78%** |
| Coal output, Mt yr$^{-1}$ | 76.50 | 72.50 | 72.20 | 70.40 | 65.50 | 63.40 |
| Specific methane emission, m$^3$ t$^{-1}$ | 11.10 | 12.30 | 12.90 | 13.30 | 14.50 | 14.40 |

a – including the methane captured by the drainage system but unused and subsequently released

## 2.1 Sources of methane emission in the area of USCB

Hard coal is a sedimentary rock of biogenic origin, containing 75-90 % of elementary carbon carbonised mainly from plant debris in the absence of oxygen. The reason for that is diagenesis, transforming the organic substances into peat and then lignite. Its furthering carbonification (metamorphism) produced hard coal and anthracite (Kotarba 1998, Czapliński 1994; Szlązak et al. 2015).

The gas emissions accompanying the extraction of coal deposits include methane, carbon dioxide, higher hydrocarbons, nitrogen, and water vapour. The released mine gas typically contains more than 86 % of $CH_4$ (Szlazak et al., 2015). The number of Polish mines in which $CH_4$ hazards occur is steadily increasing due to mining at deeper levels and continuous extraction of deposits with high methane content (Mc)(Swolkień, 2020; Szlazak et al., 2014). In contrast to specific methane emission (SME), the methane content (Mc) defines the volume of natural methane included in one tonne of dry ash-free coal (i.e. without ash and moisture content, usually given in tonne daf), and is calculated in accordance with the Polish national norm PN-G-44200 (Szlazak et al., 2013). Estimation of Mc for particular coal deposits enables forecasting of total methane emission at specific mining excavations, which in turn is needed for appropriate design of ventilation and methane drainage systems necessary to maintain safe working conditions and protect mining crews.

The methane content (Mc) in USCB varies greatly and can change within the whole USCB area and even within one coal mine deposit (Dreger, Kędzior, 2021). Additionally, it increases with depth (Kotas, 1994; Tarnowski, 1989; Dreger, Kędzior, 2021). As an example, Figure 2a shows the variability of methane content in the Brzeszcze coal mine deposit. Methane content measurements made following the procedure PN-G-44200 (Szlazak et al., 2013) in selected excavations (marked in the diagram with white dots) enabled researchers to draw a Mc isoline map in a two-dimensional system. Additionally, the map shows geological disturbances in the form of faults and anticlines, which can significantly increase the value of methane content in the deposit. It can vary between 4 to even 16 m$^3$ $CH_4$ t$^{-1}$ daf, and its increased values are visible in places of geological disturbance. Figure 2b, on the other hand, shows the distribution of methane content relative to depth in the form of a frame plot for the particular depth intervals in the Rydułtowy coal mine. We can see that at the depth of -1000 m below msl (level 1200 m), the average methane content increases to 5.0 m$^3$ $CH_4$ t$^{-1}$ daf, with determined methane content being in the range from 1.5 to 11.4 m$^3$ $CH_4$ t$^{-1}$ daf.

Currently, in Polish mines, coal is extracted at an average depth of 800 meters below msl. However, in many of the plants, extraction is conducted at depths exceeding 1000 m below msl. As a result, the deeper the coal mines go into exploitation, the greater the amount of released methane, which is reflected by WUG data presented in Table 1.

The numbers show that, despite the decreasing number of active underground coal mines in Poland and reduced coal output, the specific methane emission increased between 2013 and 2018 (see Table 1).

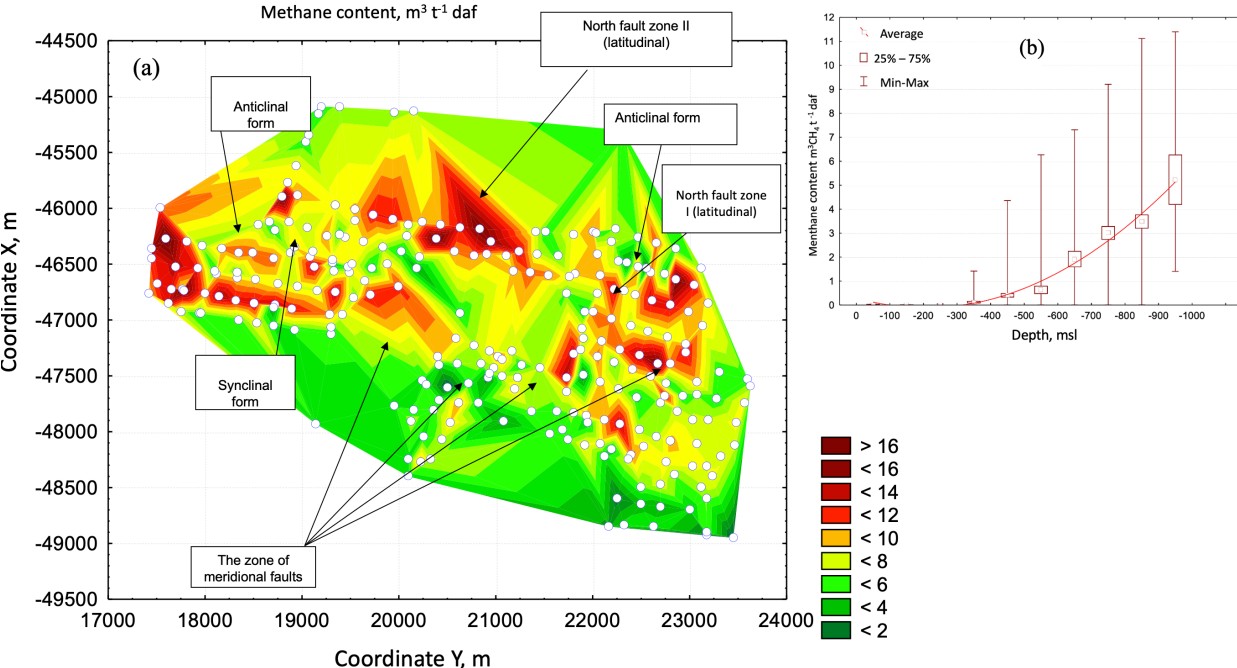

**Figure 2.** Methane content (Mc) variability depending on depth and across the deposit exemplarily shown for the Brzeszcze coal mine. (2a) Wafer chart of the methane content isolines within the deposit 352 in Brzeszcze coal mine in two-dimensional space with marked disturbances of the geological structure. White dots indicate methane content measurement points. The disorders observed based on the geological structure of the deposit are described in square boxes. X and Y are the coordinates. (2b) Frame chart of methane content variability at individual depth intervals in Rydułtowy coal mine with significant mean values as a midpoint. The box corresponds to the range of results from 25% to 75%, while the whiskers represent the minimum and maximum values.

## 2.2 Methods of methane capture

For the sake of the safety of mining crews working underground, hard coal mines in Poland and worldwide have to employ methane prevention methods (Journal of Laws 2017, item 1118) to ensure adequately low methane concentrations in mining excavations. Coal mines use active ventilation systems depending on the forecasted total methane emission, and a drainage system can sometimes accompany them. Both installations must be designed specifically for a particular excavation site (longwall). Proper ventilation involves supplying sufficient air to each excavation in the mine to guarantee safe $CH_4$ concentration, i.e. lower than 2 % (Journal of Laws 2017, item 1118). Very often, low $CH_4$ emission means that drainage is technically challenging and economically infeasible (Swolkień, 2020), which usually occurs when the forecasted total methane emission for the particular longwall is below 10 $m^3$ $min^{-1}$ (Swolkień, 2020). In that case, $CH_4$ is removed directly into the atmosphere using a ventilation system, only. On the other hand, when the longwall emissions are above 10 $m^3$ $min^{-1}$, the air supply to the excavation is generally insufficient to reduce $CH_4$ concentration to a safe level. In such a case, the coal mines employ the second method, i.e., methane drainage (Journal of Laws 2017, item 1118). A properly designed drainage system reduces the ventilation air

methane in excavations and the frequency of methane inflows into operating areas. It also prevents or reduces events such as outflows and abrupt outbursts of methane and rocks.

Hard coal mines around the world use different methods of methane drainage. One of the most widespread, pre-mining methane drainage, is mainly used in the United States and Australia (Fields et al., 1973; Daimond, 1994; Kissel, 2006; Schatzel et al., 2008; USEPA, 2009; Black, Aziz, 2009). It involves capturing methane for up to several years before coal extraction operations begin. Mining methane drainage, used mainly in Poland, Ukraine and Russia, has been described in numerous publications (Shirin et al., 2011; Szlązak, Swolkień, Borowski, 2014, Szlązak,

Swolkień, Obracaj 2014; Szlązak et al., 2015; Swolkień 2015; Leisle, Kovalski, 2017). While specific applications differ from country to country, the general principle of methane capturing consists in draining it from the rock-mass and isolating goafs through specially designed boreholes. Later, the gas is discharged via a separate system of pipelines onto the surface, using the low pressure generated in a methane drainage station. The parameters and placement of the drainage boreholes depend on the ventilation system and the local conditions related to geology and mining activities.

The decisive factor determining methane capture and, therefore, the efficiency of methane drainage is the large number of boreholes connected to the drainage system with negative pressure (in front of and behind the face). In other words, a drainage system that can ensure high efficiency of methane capture will cause a decrease in its emission to the excavation and then to the ventilation shaft.

       Worldwide, currently implemented technologies enable capturing methane from particular longwalls with an

efficiency ranging between 70 % and 80 %, depending on the forecasted total methane emission (Fields et al., 1973; Daimond, 1994; USEPA, 2009; Swolkień, Szlazak, Obracaj, 2014; Szlązak et al., 2015). That, in turn, means that 20 % to 30 % of methane is released as ventilation air methane (VAM). Currently, the total drainage efficiency of capturing $CH_4$ in Poland is, on average, only 35 % (see Table 1). The ventilation shafts release the remaining 65 % directly into the atmosphere.

The coal mine plants in Poland utilise drained methane internally or sell it to external power plants (Szlązak et al., 2014; Swolkień, 2015). The functioning of the vast majority of such plants is based on internal combustion engines with pistons, because of their high efficiency and relatively small investments needed. Notably, the captured methane is utilised almost exclusively in winter (during the heating season). In summer, it is typically released into the atmosphere or flared. Unfortunately, there is no publicly available data specifying the share of methane processed

using each of the two methods.

       The average methane utilisation efficiency in Polish coal mines, computed for drained methane, is about 63 %. What follows is that 37 % of the methane is released from the coal mine drainage station, usually located inside the primary mine compound on the surface. Therefore, coal mine drainage stations should be treated as additional methane emission point sources producing non-negligible emissions, as the amount of methane emitted from them

reach on average 13% of total annual emissions (see Table 1).

       In order to assess the variability of that average at the plant level, we have gathered data on the annual utilisation of drained gas in a subset of mines targeted in Comet 1.0 (see section 3). Over the whole of 2018, the efficiency of drained methane utilisation for all the plants of the JSW S.A. company reached 57%, of which 75% was sold to external power plants, 24 % was burnt in gas engines, and ca. 1% was utilised in gas boilers (Szlązak, Swolkień,

2021). The highest consumption of drained methane, reaching 83 %, was achieved by the Zofiówka, Borynia, and Pniówek plants. On the contrary, Szczygłowice and Budryk utilised the least methane, 5 % and 40 %, respectively. In the case of Knurów, it released all (100%) of its drained methane into the atmosphere.

The above examples show that utilisation of drained methane is highly variable, and significant errors in the emission estimates can thus occur at the plant level to inform global databases.

Unfortunately, data about the utilisation of drained methane is proprietary and not publicly available. Until such information is publicly released, any efforts to quantify methane emitted from drainage installation are bound to be limited in accuracy and precision. Furthermore, attempting to estimate emissions on sub-annual temporal scales will be even more problematic, as the high-frequency monitoring data is either hard to obtain or do not exist.

In order to compare the atmospheric methane emission from the USCB mining areas with other mining regions in Poland, it is necessary to consider the prevailing mining and geological conditions and the methane drainage methods employed (Swolkień, 2020). The considerations presented in this work concern mining areas where exploration is carried out at considerable depths. Additionally, the rock-mass is characterised by high methane content increasing with depth (see Fig. 3a, b), low permeability, and high ground stress (Szlązak et al., 2012; Roszkowski, Szlązak 1999; Szlązak et al., 2015; Szlązak, Borowski 2006). A similar situation occurs in most coal mines in Ukraine, Russia and China (Zahai et al., 2008; Romeo, 2013; Younn et al., 2012; Boger et al., 2014; Szlazak et al., 2012). The shallower coal deposits are characterised by higher permeability, as a result of which it is possible to apply pre-mining drainage (Karacan et al., 2011; Greedy, Tilley, 2003). Ideal conditions exist in mines in the United States, although this method is also applied in Australia (Kissel, 2006; Daimond, 1994; Fields et al., 1973; USEPA, 2009; Schatzel et al., 2008; Black, Aziz, 2009).

Because, as of 2019, Germany has closed its coal mines, and the Czech Republic will do so by the end of 2022, the USCB in Poland was the best showcase to conduct measurements of coal mine methane during the CoMet mission.

### 3        The USCB as a case study for CoMet 1.0 mission

Because the USCB is an area characterized by highly concentrated $CH_4$ emissions, it is ideal for comparing all scales of surveys: ground-based, airborne, and satellite. The measurements carried out during the pre-campaign in 2017 (CoMet 0.5) were the first attempts to observe the temporal emissions of $CH_4$ directly from the source (Swolkień, 2020; Andersen et al., 2021). Until then, only ground-based measurements were carried out around the shafts (Nęcki et al., 2017). The absence of information about specific temporal concentrations that influence factors is a significant obstacle to validating actual emission rates from aircraft measurements (in-situ, remote sensing). Data collected during the pre-campaign were also beneficial to planning and prioritising the USCB mines for the primary CoMet 1.0 mission, which took place from May 14 to June 13, 2018 (Swolkień, 2020; Kostinek et al., 2020; Fiehn et al., 2020; Luther et al., 2019; Krautwurst et al., 2021). The preliminary study led to the selection of seven coal mine facilities with 15 ventilation shafts in Upper Silesia for the investigation within CoMet 1.0. Figure 1 presents the map of the mining areas of the USCB indicating the mines with verified methane release and also ventilation shafts.

The CoMet 1.0 campaign, along with the observations and tests conducted from aircraft and on the ground, performed measurements of methane concentrations directly in the ventilation shafts of selected individual coal mine

facilities. The research covered all coal mines that belong to the JSW S.A. company in an area covering approximately 195,3 km$^2$:

- The Pniówek coal mine – three ventilation shafts.
- The combined entity Zofiówka-Borynia-Jastrzębie consisting of three individual facilities – six ventilation shafts.
- The Budryk coal mine – two ventilation shafts.
- The combined entity Knurów-Szczygłowice consisting of two individual facilities – four ventilation shafts.

JSW S.A. is the largest coking coal producer in the European Union and one of the leading producers of coke, which is an essential ingredient for steel production. In 2018, its coal mines released into the atmosphere 237,77 kt of CH$_4$ and were responsible for 16 % of total atmospheric emissions from the underground sector in Europe (Szlązak, Swolkień, 2021). An advantage of selecting these mines as mission targets was their localisation and the fact that they are the most methane-prone.

**4        The methodology of methane concentrations and emissions measurements**

According to the regulation (Directive 2003/87/E.C. December 19, 2018; Journal of Laws 2016 item 1877), all industrial companies in Poland, including coal mines, are obliged to report greenhouse gas emissions to the National Centre for Emissions Management (KOBiZE- IOŚ-PIB). KOBiZE monitors fulfilling commitments imposed by the EU Directives and the EU Emission Trading System (EU ETS). In addition, coal mines are obliged to report emissions to the National Pollutant Release and Transfer Register (The Environmental Protection Low of April 27, 2007, art. 236a) on an annual basis.

Measurement of methane emissions for the purpose of emission reporting employs a combination of measurements and calculation methods. As high-precision methane measurement devices are not required for regular mining operations, emission reporting relies on a limited number of samples collected across the mine that are subsequently measured in a certified laboratory. According to the law, these measurements are only required to be performed at a limited frequency (monthly). Due to the fact that the air flows in the vertically oriented ventilation shafts can be extremely large (up to 22 000 m3 min-1), it is impractical to collect samples directly in the main exhaust shaft. Instead, the collection of samples (into special vacuum bottles known as "Gresham tubes") is usually performed by a trained employee at the intersection between horizontal return airways at the lower levels of the mine (return shafts), also equipped with a hand-held anemometer to measure the air flows reliably. After the methane concentrations in the collected samples become available, the emissions are estimated by calculating the average concentration across the return shafts (weighted by the respective air flows), using the following formula:

$$Q_{CH4} = Q_{air} \cdot {}^{c_{CH4}}\!/_{100} \tag{1}$$

where $c_{CH4}$ is methane concentration (in % vol), and $Q_{air}$ is the air flux in m$^3$ min$^{-1}$. This averaged concentration is subsequently multiplied by the total air flow measured at the output of the ventilation shaft, thus providing the emission estimate.

Frequent (up to hourly) concentration measurements would be a helpful tool to determine methane emissions in coal mine shafts. All coal mines, including the ones described in the paper, are obliged to employ the methane fire

teletransmission monitoring system to monitor safety parameters in accordance with the Polish mining regulations (Journal of Laws 2017 item 1118). These parameters usually include, among other things, methane, carbon monoxide and oxygen concentration, air velocity, temperature, and pressure. Although the safety system measures methane concentrations and air flows over multiple locations below ground (including return shafts), coal mine operators rarely use it to estimate methane emissions to the atmosphere. In Poland, only the Pniówek coal mine reports emissions based on the indications of the monitoring system; all the remaining mines use the monthly sample-based estimations, as described earlier.

The most widely used system in Polish coal mines is the SMP-NT/A (Polish: "*System Metanowo - Pożarowy*") with integrated CMC-3MS telemetry panels (called CTMs, from Polish: "*Centrala TeleMetryczna*") (Wojaczek, Wojaczek, 2017; emagserwis.pl; Wasilewski, 2012; Swolkień, Szlązak, 2021). Figure 3 shows a diagram of an SMP-NT/A system. In such systems, each methane sensor, equipped with a continuous recording feature, protects all ongoing faces, longwalls and ventilation shafts and is usually connected directly to a telemetry panel via an intrinsically secure telecommunication network that allows sending data to and from the sensors to the CTM. All sensors connected to the monitoring system protect longwalls and roadways in use, both at the inlet and the outlet. Their primary role is to control methane concentration; in case of exceedances, they are equipped with contacts for switching off the electricity. The mine monitoring system is pervasive, and the number of methane sensors in one mine can reach up to 200. Methane sensors constitute 60 % of all sensors installed in coal mines. Other sensors (e.g. $CO_2$, $O_2$), due to lower electricity consumption, can be connected to underground pit stations (S.D.). The transmission line from the CTM switchboard on the surface energises an underground station. Contemporary environmental parameter control systems monitor hazards (methane and other ventilation parameters), but most of all, they perform safety-related functions, such as switching off electricity in the endangered area once the threshold settings of the sensor are exceeded, as well as displaying local alarms about exceeding the threshold settings in the place where the sensor is installed (Wojaczek, Wojaczek, 2017).

The methane measuring apparatus consists of a DCH recording sensor (type MM-4), shown in Figure 3 (emagservice.pl; Swolkień 2020). Continuous monitoring of $CH_4$ concentration in the surrounding air is achieved with the use of two independent elements covering two $CH_4$ concentration ranges: i) a pellistor gas detector for low range (0–5 % vol) and a conductivity bridge for high range (5–100 % vol). Sensors can be switched automatically, and their heads can be placed on the cover or through an up to 30-m long cable. A replaceable filter protects the air inlet to the sensor head. The sensor's response time is less than 6 seconds, it can operate in temperatures ranging from -10 ºC to 40 ºC and in relative humidity ranging from 0 to 95 %. The nominal resolution of methane measurement is 0.1 %, with absolute measurement uncertainty of 0.1 % vol at the low concentration range and 3 % vol at high concentration range (emagserwis.pl).

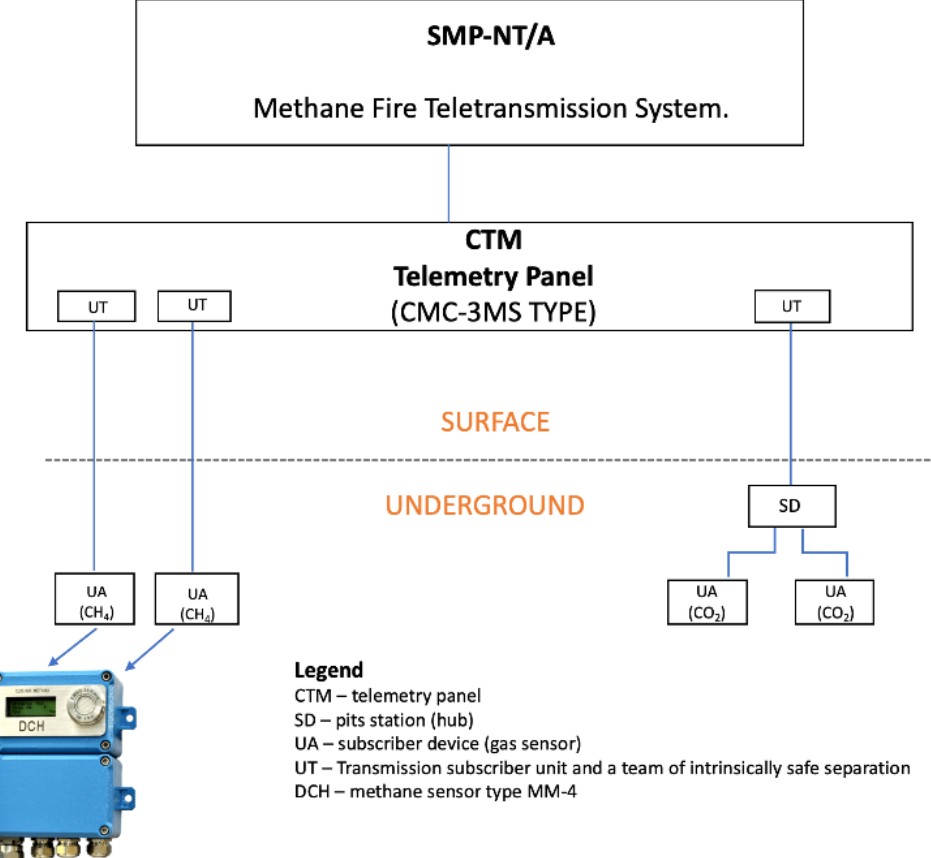


**Figure 3.** Block diagram of a system for monitoring environmental parameters with a star-shaped structure of the teletransmission network, frequently employed in Polish coal mines, including commonly used methane sensor type MM-4 (production EMAG Serwis sp. z o.o.; emagserwis.pl)

The European standard IEC 60079-0:2013-03 sets out the requirements that methane sensors have to fulfil, including their maintenance and calibration methods. This regulation gives guidance and recommends practice for the selection, installation, safe use, and care of electrically operated Group II equipment intended for use in industrial and commercial safety applications and Group I equipment in underground coal mines for the detection and measurement of flammable gases (PN-EN 60079-0:2013-03; PN-EN IEC 60079-0:2018-09).

Calibration of the measurement sensors in the ventilation shafts consists of two steps:

- once a week with the mixture of 2.2 % vol of methane for a low concentration range (0-5 % vol),
- every two weeks with a blend of 70 % vol methane (0-100 % vol) for a higher concentration range.

Figure 4 presents the ventilation shaft scheme, with black dots indicating the measuring sensor locations. Methane sensors are placed at the edge of the ventilation channel, approximately 10–15 m below the inlets. The air

temperature in the shaft is approximately 18 to 20 °C, and the height of the ventilation tower is about 10 to 15 meters.

According to Polish mining regulations (Journal of Laws 2017 item 1118), the methane concentration in the collective airflows of the return air must not exceed 0.75 %, vol and the corresponding measurements must be performed in the ventilation shaft in the joint return airflow, not less than 10 m:

- below the channel of the main fan (see Fig. 4, black dot),

• above the highest return air inlet flowing from the excavations to the ventilation shaft.

Additionally, stations of the main fan should consist of instruments that perform continuous measurements of:

• the static air pressure in the ventilation channel in front of and behind the main valve (see Fig. 4, red dots),

• the air velocity in the ventilation channel (see Fig. 4, blue dot),

• the static air pressure in the cross-section of the exhaust shaft below the ventilation channel.

The methane sensors described above are part of the SMP-NT/A monitoring system and are used in mines as devices to control whether methane concentrations do not exceed the legal value of 0.75 % vol. Their application implies large uncertainty of methane emissions. Due to that, they are not designed to measure methane concentration for the sake of reporting emissions. It is possible to use other sensors, e.g. TDLAS (tunable laser diode absorption spectrometer) analyser, directly over the ventilation shaft diffuser. However, to our knowledge, coal mines have never

used such a solution. Therefore, it is not possible to say whether they would be able to operate correctly in the supersaturation conditions of the upper parts of the ventilation shafts. It is possible to use both open path and closed path TDLAS instruments. The open path instrument measures the averaged methane concentration at the shaft cross-section. In contrast, the closed path analyser measures the methane concentration at a single point at the exhaust of the ventilation shaft. The second one can be potentially installed at different locations, also inside the shaft – informing

about the homogeneity of the air stream. The possibility of using the sensors mentioned above in the ventilation shafts of selected mines in USCB is the subject of a preliminary research project currently being proceeded by the International Methane Emissions Observatory (IMEO).

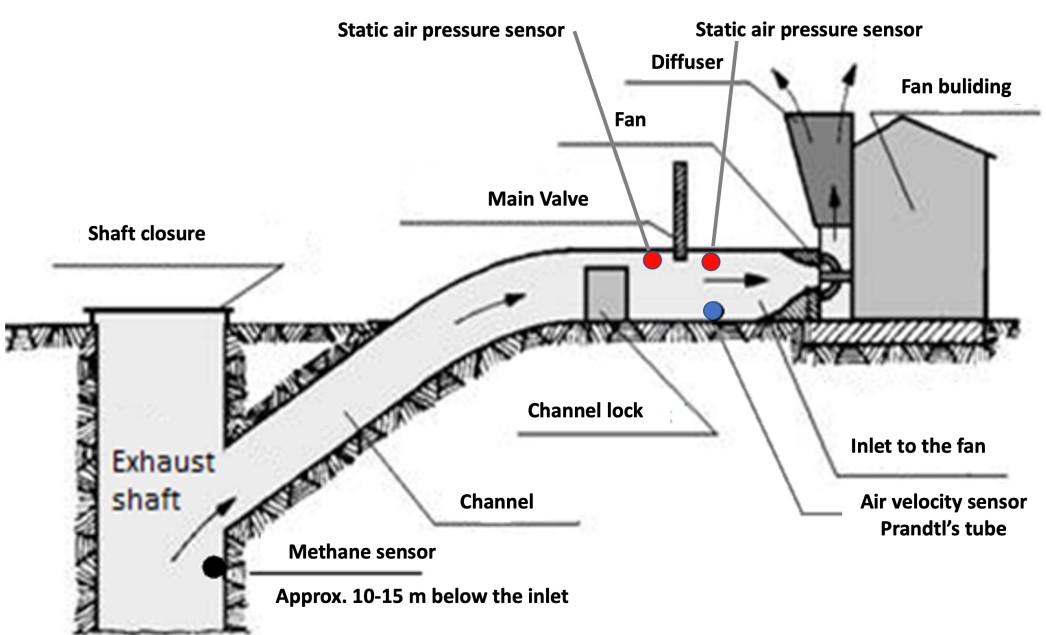

**Figure 4.** Scheme of a typical ventilation shaft scheme where colour dots stand for methane, velocity, and static pressure sensors; quoted after Swolkień 2020. The original Figure was published under a Creative Commons Attribution 4.0 International 225 License, http://creativecommons.org/licenses/by/4.0/.

Usually, fan stations and diffusers are designed individually for each coal mine, depending on the scope of mining works performed, and the magnitude of air flux supplied to the respective mining areas. Nevertheless, the general principle of their construction is similar for all coal mines. Figure 5a presents the top view of the diffusers (air exhaust), and Figure 5b shows the connection of the ventilation channel with the diffuser through the channel lock and main valve (the Bogdanka coal mine, www.stalkowent.pl). The last photograph (Fig. 5c) presents the main valve and the fan (the view from inside).

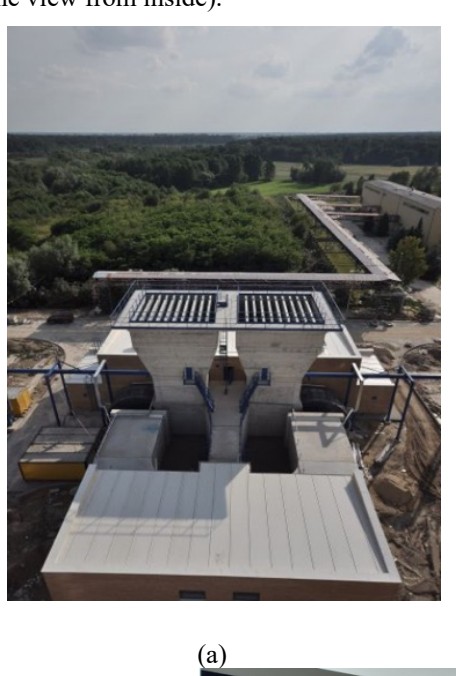
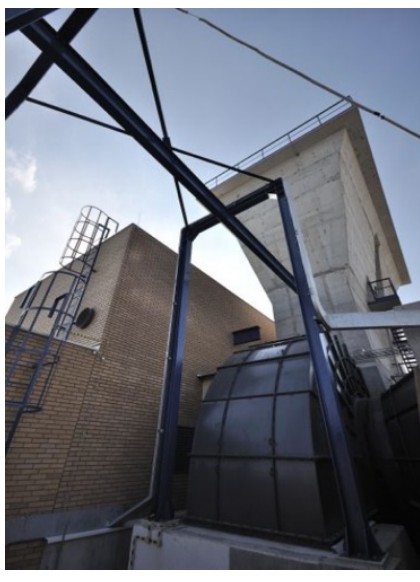

(a)

(b)

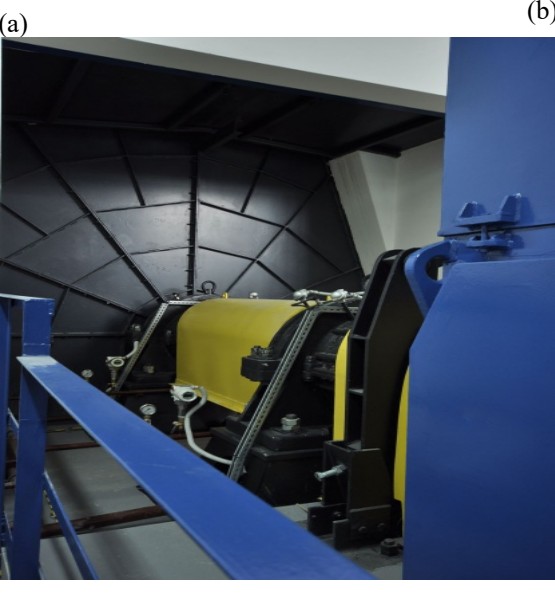

©


**Figure 5**. Photographs of the fan station with diffuser in the Bogdanka coal mine built by Stalkowent sp. z o.o: (a) A fan station with a diffuser – view from above; (b) A photograph of the channel lock, the main valve, and the diffuser; (c) The main valve and the fan. Source: promotional materials of Stalkowent sp. z o.o.: https://www.wnp.pl/gornictwo/stacja-wentylatorow-dla-lw-bogdanka-sa-wykonana-przez-stalkowent-sp-z-o-o,-7642.html.


In order to quantify the methane flow rate, it is necessary to measure the air velocity in the ventilation channel. A Prandtl tube between the main valve and the fan (see Figure 4, blue dot) connected with a manometer continuously records dynamic pressure (Swolkień 2020). The airflow rate through a ventilation channel is calculated with the formula

$$Q_{air} = 60 \cdot A_{ch} \cdot v_{ch} \qquad (2)$$

where $Q_{air}$ is the resulting air flow rate (in $m^3$ $min^{-1}$), $A_{ch}$ is the channel cross-section ( in $m^2$), and $v_{ch}$ is the air velocity (in m $s^{-1}$) measured as described above. The methane flow rate (in $m^3CH_4$ $min^{-1}$) is then calculated with the formula

$$Q_{CH4} = 0.95 \cdot Q_{air} \cdot {}^{c_{CH4}}\!/_{100} \qquad (3)$$

where $c_{CH4}$ is methane concentration (in % vol). The index of 0.95 results from the fact that 5% of the air discharged
through the shaft comes from the shaft closure (see Fig. 4), which must be considered when calculating the methane flow rate. The next step is the conversion into methane emission rate, which is done with the formula
$$m_{CH4} = Q_{CH4} \cdot \rho \qquad (4)$$
where $\rho$ is methane density (0.717 kg $m^{-3}$) referred to normal conditions.

Regardless of the adopted methodology for measuring the total methane emissions from individual coal mine
facilities (STMP-NT/A or traditional methodology), the amount of methane released from the ventilation shafts is always increased by the amount of methane blown out into the atmosphere at the methane drainage station. Therefore, the last parameter is measured using the thermal flow meter installed in the discharge pipeline and by measuring methane concentration with a chromatograph.

The relative uncertainty of the airflow rate measurements usually amounts to ~10 %; for methane concentration
absolute measurement uncertainty amounts to ~0.1% vol. Thus, the relative uncertainty of the flux should be dominated by the uncertainty of the airflow. However, harsh conditions that occur in the ventilation shaft, namely high humidity and waterlogging, often force the measuring equipment in the shafts to operate outside of the nominal operating range. Considering that, we estimate the overall methane flux uncertainty calculated with this method to be higher than the respective instrument uncertainties would indicate. Therefore, we consider 20 % as an appropriate
conservative estimate of the uncertainty of the estimated methane emission.

The analysis of the coefficients influencing the variability of methane emissions presented in the article was based on instantaneous measurements of methane concentrations using the methane fire teletransmission monitoring system (SMP-NT/A) and methane sensors installed in the ventilation shafts of the studied mines (see Fig. 1).

5    Factors that influence the methane emissions from coal mines

5.1. Results from in-stack measurements
Figures 6a and 6c depict variations in the concentrations and methane fluxes in the selected 15 ventilation shafts under study, based on hourly values. Figure 6b presents an overview of average air flow rates in each ventilation shaft.

All temporal data for individual shafts of selected coal mine facilities presented below, together with emissions data
from the State Mining Authority (WUG, 2019) and E-PRTR (E-PRTR), were compiled by the CoMet team in the
form of an internal dataset called CoMet v4.0 (CoMet E.D. v4, 2021). Additionally, Tables 2 and 3 present results of
statistical description of concentration and flow rate variations.

The analysis of data presented in Figure 6a and Table 2 shows that the highest average values of methane
concentrations were recorded for Budryk V (0.40 %), followed by Pniówek IV (0.26 %). The lowest values, on the
other hand, were observed at Knurów Bojków V, Borynia III, and Jastrzębie IV shafts. We find a striking variability
of methane concentration in the shafts mentioned above. Despite the low average concentration, the highest recorded
values reached up to 0.13 % at Borynia III and 0.30 % at Knurów Bojków V. Similarly, in Budryk II and Pniówek V,
concentrations varied between 0 and 0.46 %. It should be noted that in Borynia VI, in the period from May 17 to 30,
methane sensors displayed zero concentration which is impossible since this shaft is responsible for ventilating the
main mining areas of the plant. According to the source file, it malfunctioned during the indicated period.
Consequently, this period was excluded from further calculations.

The data presented in Figure 6a shows that in none of the monitored ventilation shafts was the maximum
permissible value of methane concentration of 0.75% exceeded (Journal of Laws 2017 item 1118). Nevertheless,
because of the high temporary variability of methane concentrations in individual shafts, their instantaneous values
were 1.68 (Borynia VI) and 1.91 (Budryk II) times higher than the average for the analysed period. In the case of
Knurów Bojków V, this deviation was even higher. Large differences in concentrations can cause artificial over- and
underestimations of flux when data from atmospheric measurements (e.g. from devices installed onboard aircraft) are
used to infer emissions. This is because the observational window of the airborne measurements is usually limited to
two or three hours for any given source, and sometimes even shorter. This temporal variability of emissions will affect
primarily the measurements performed in the immediate vicinity of the sources, as further downwind the atmospheric
mixing in the atmosphere should allow measurements of an averaged signal. Particularly dangerous in this regard are
cases where rapid and persistent changes of methane emissions from specific shafts occur. Without accurate readings
of temporal methane concentrations in the ventilation shafts, even accurate calculation of momentary fluxes will lead
to discrepancy between estimated values and annual reported emissions.

The data presented in Figure 6b reveals that the average values of air flux in the shafts ranged from 6000 $m^3$ $min^{-1}$ for Knurów Aniołki to over 23 000 $m^3$ $min^{-1}$ for Zofiówka IV. The magnitude of air flux depends on the scale of
mining works performed in the individual coal mine. It means that the larger the coal mine (more longwall panels with
high methane emissions), the higher the air flux. It is crucial to remember that the air flux for each ventilation shaft is
set based on the number of longwalls under operation in a given facility to maintain safe operating conditions
underground. The last column in Table 3 contains information about the frequency of air flux measurements. Some
results are hourly, and some are averaged over a day or a month. The value of air flux varies from shaft to shaft but is
relatively stable across the month under study (Fig. 6b). It can, however, change slightly once the scale of extraction
is reduced (removal of one or more longwalls), which nevertheless requires substantial changes to the ventilation
system of the mine.


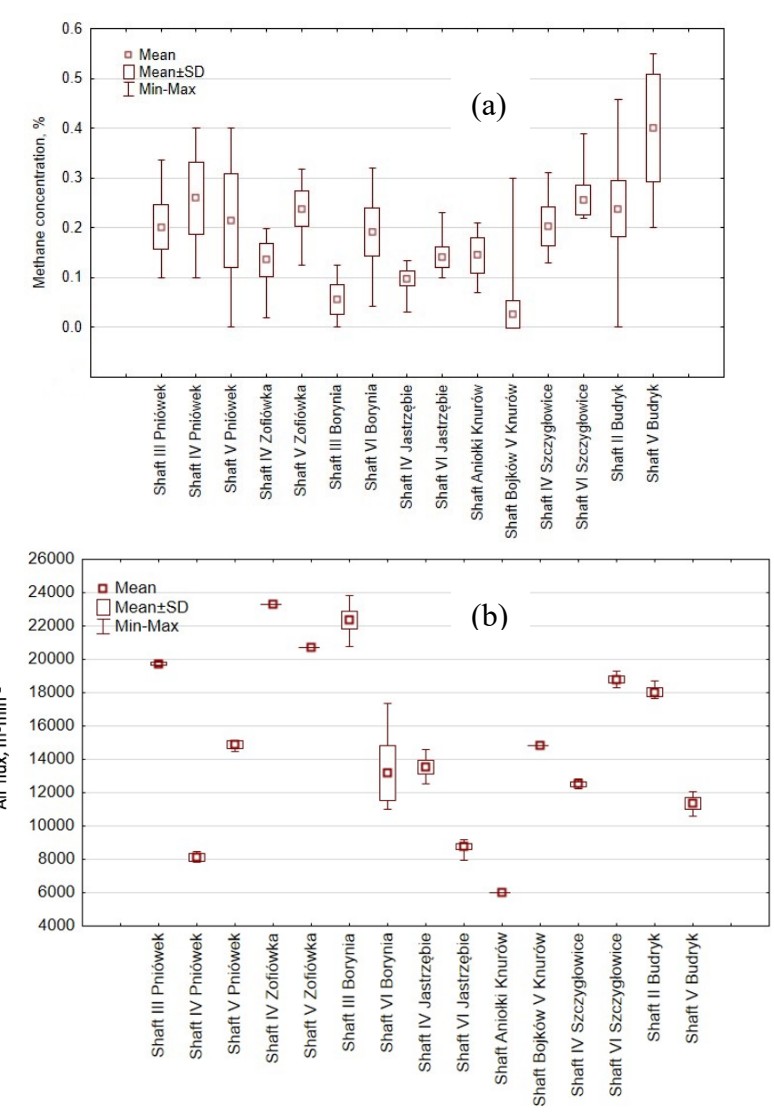

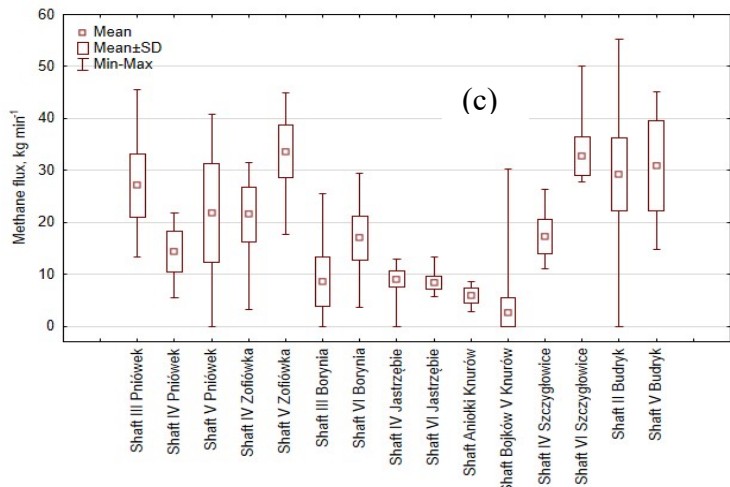

**Figure 6**. Changes in methane concentration (6a), air flux (6b) and methane flux (6c) in the investigated shafts within the CoMet 1.0 observation period from May 14 to June 13, 2018

**Table 2**. The statistical description of methane concentration variation in ventilation shafts under study (period from May 14 to June 13, 2018)

| Statistical description (CH₄ concentration variation), % | | | | | | |
|---|---|---|---|---|---|---|
| Name of the shaft | Average | Median | Min. | Max. | Standard deviation | Frequency of concentration data |
| Shaft III Pniówek | 0.20 | 0.20 | 0.10 | 0.34 | 0.05 | hourly |
| Shaft IV Pniówek | 0.26 | 0.30 | 0.10 | 0.40 | 0.07 | hourly |
| Shaft V Pniówek | 0.22 | 0.20 | 0.00 | 0.40 | 0.09 | hourly |
| Shaft IV Zofiówka | 0.14 | 0.14 | 0.02 | 0.20 | 0.03 | hourly |
| Shaft V Zofiówka | 0.24 | 0.24 | 0.13 | 0.32 | 0.04 | hourly |
| Shaft III Borynia | 0.06 | 0.04 | 0.00 | 0.13 | 0.03 | hourly |
| Shaft VI Borynia | 0.19 | 0.17 | 0.04 | 0.32 | 0.05 | hourly |
| Shaft IV Jastrzębie | 0.10 | 0.10 | 0.03 | 0.14 | 0.02 | hourly |
| Shaft VI Jastrzębie | 0.14 | 0.15 | 0.10 | 0.23 | 0.02 | hourly |
| Shaft Aniołki Knurów | 0.15 | 0.14 | 0.07 | 0.21 | 0.04 | hourly |
| Shaft Bojków V Knurów | 0.03 | 0.02 | 0.01 | 0.30 | 0.03 | hourly |
| Shaft IV Szczygłowice | 0.20 | 0.21 | 0.13 | 0.31 | 0.04 | daily average |
| Shaft VI Szczygłowice | 0.26 | 0.25 | 0.22 | 0.39 | 0.03 | daily average |
| Shaft II Budryk | 0.24 | 0.25 | 0.00 | 0.46 | 0.06 | hourly |
| Shaft V Budryk | 0.40 | 0.45 | 0.20 | 0.55 | 0.11 | hourly |

When it comes to methane fluxes, the variability of concentrations transfers to the fluxes and is most visible in Budryk II and V, Knurów Bojków V, Borynia III, and also Pniówek V (see Fig. 6c). The maximum flux for Knurów Bojków V was 11.60 times the average value and for Borynia III it was almost threefold. The largest amount of methane was released from Zofiówka V and Szczygłowice VI, followed by Budryk V and II, while the smallest amount

was released from Knurów Bojków V and Knurów Aniołki (see Table 3).

Obviously, the higher the air flux is, the more diluted the gas. It becomes clear once methane concentrations and fluxes in both shafts are compared. The values of methane concentration were more or less the same, $0.15 \pm 0.1$ % for Knurów Aniołki and $0.14 \pm 0.1$ % for Zofiówka IV, respectively (see Table 2), while flux in the second-mentioned shaft was 3.6 times the one from Knurów Aniołki (see Table 3 and Figure 6b). A similar situation existed also in Budryk V, characterised by the highest concentration from all shafts, $0.40 \pm 0.1$ % (see Table 2), but releasing less methane than Zofiówka V and Szczygłowice VI (see Table 3). As seen in Figure 6b, air flux is stable in all ventilation shafts, which means that everything that happens inside the mining areas ventilated by an individual shaft is reflected in the value of the concentrations. These changes are then transferred to the emissions.

**Table 3**. The statistical description of methane fluxes calculated for ventilation shafts analysed in the study (for period May 14th to June 13th, 2018). Frequency of the flux data varies according to the frequency of base data provided from administration of the respective mines.

| Statistical description (CH₄ flux variation), kg min⁻¹ | | | | | | |
|---|---|---|---|---|---|---|
| **Name of the shaft** | **Average** | **Median** | **Min.** | **Max.** | **Standard deviation** | **Frequency of air flux data** |
| Shaft III Pniówek | 27.08 | 26.84 | 13.42 | 45.54 | 6.08 | hourly |
| Shaft IV Pniówek | 14.32 | 15.94 | 5.52 | 21.80 | 3.93 | hourly |
| Shaft V Pniówek | 21.79 | 20.43 | 0.00 | 40.87 | 9.50 | hourly |
| Shaft IV Zofiówka | 21.51 | 22.50 | 3.18 | 31.50 | 5.29 | monthly average |
| Shaft V Zofiówka | 33.64 | 33.86 | 17.67 | 44.88 | 5.07 | monthly average |
| Shaft III Borynia | 8.59 | 5.94 | 0.00 | 25.49 | 4.75 | hourly |
| Shaft VI Borynia | 16.98 | 16.78 | 3.56 | 29.51 | 4.24 | hourly |
| Shaft IV Jastrzębie | 9.09 | 8.97 | 0.00 | 12.88 | 1.59 | hourly |
| Shaft VI Jastrzębie | 8.38 | 8.85 | 5.78 | 13.43 | 1.28 | hourly |
| Shaft Aniołki Knurów | 5.93 | 5.70 | 2.84 | 8.58 | 1.45 | monthly average |
| Shaft Bojków V Knurów | 2.61 | 2.02 | 1.01 | 30.24 | 2.77 | monthly average |
| Shaft IV Szczygłowice | 17.27 | 17.51 | 11.01 | 26.34 | 3.26 | daily average |
| Shaft VI Szczygłowice | 32.71 | 32.18 | 27.85 | 50.01 | 3.67 | daily average |
| Shaft II Budryk | 29.29 | 30.28 | 0.00 | 55.35 | 7.03 | daily average |
| Shaft V Budryk | 30.98 | 33.34 | 14.82 | 45.18 | 8.67 | daily average |

### 5.2. Tentative reasons for the findings

The explanation for significant fluctuations in methane concentrations and fluxes presented in Figures 6a and 6c is the release of methane from rock-mass caused by mining activity. The rock-mass is a porous medium, and the methane flows through interconnected spaces and channels or tiny fractures due to the filtration process. In fact, in underground conditions, filtration is a very complex process since the coal seams are not only the rock through which the methane filters, but also contain methane in adsorbed form (Swolkień, 2015). The phenomena of desorption and simultaneous filtration are closely related mechanically and energetically.

The potential of methane migration depends on the permeability of the deposit and its saturation with methane (for methane content, see Fig.2a & b), which means that variations in the composition and structure of rock-mass

influence the level of CH4 concentrations and their fluxes. Therefore, the methane emission rate from the rock-mass will depend on its permeability and, to a large extent, may vary depending on the mining region under analysis. The process of methane filtration from the coalbed to the excavation occurs due to rock depressurising caused by mining, which lowers the pressure of free gas and contributes to its desorption.

Under the conditions of low permeability, methane emission occurs only after the decompression of the rock-mass and strongly depends on the scope of mining work. Its value is the highest at the longwall face (between 0 and 20 meters) and decreases as the distance from the face grows. Then it stabilises at a level corresponding to the conditions in an unextracted seam, and methane emission decreases. Consequently, methane can be emitted to mining excavations due to its desorption and gradual filtration under the pressure gradient caused by mining or as a result of

its outflow from fractures and cracks in the seam caused by the mining operations (Szlązak et al., 2015). An increase in mining activity will always result in methane outflow, regardless of the form the outflow takes.

The fluctuating values recorded in the Pniówek V and IV coal mine shafts are striking evidence of the complexity of the processes that accompany mining activity. Figure 7 presents the changes in methane concentration in those shafts during the observation period. The vast majority of recorded results range from 0.10 to 0.40 % ± 0.1 %. In

shaft V, the concentration values ranged from 0.10 to 0.40 % ± 0.1 % during the first half of the research period. After May 31, a decrease was observed, with the recorded values ranging from above zero (except for ten cases in which zero values were identified) to 0.20 % ± 0.1 %. In the case of shaft IV, the values ranged from 0.10 to 0.30 % ± 0.1 % until May 18. Then they dropped and remained at a level of 0.10 to 0.20 ± 0.1 % until May 28. Afterwards, the range changed from 0.20 to 0.40 ± 0.1 %. In the case of Shaft III, the values fluctuated between 0.10 to 0.20 %. ± 0.1

%. This range shifted upwards (from 0.20 to 0.30 % ± 0.1 %) after June 8.

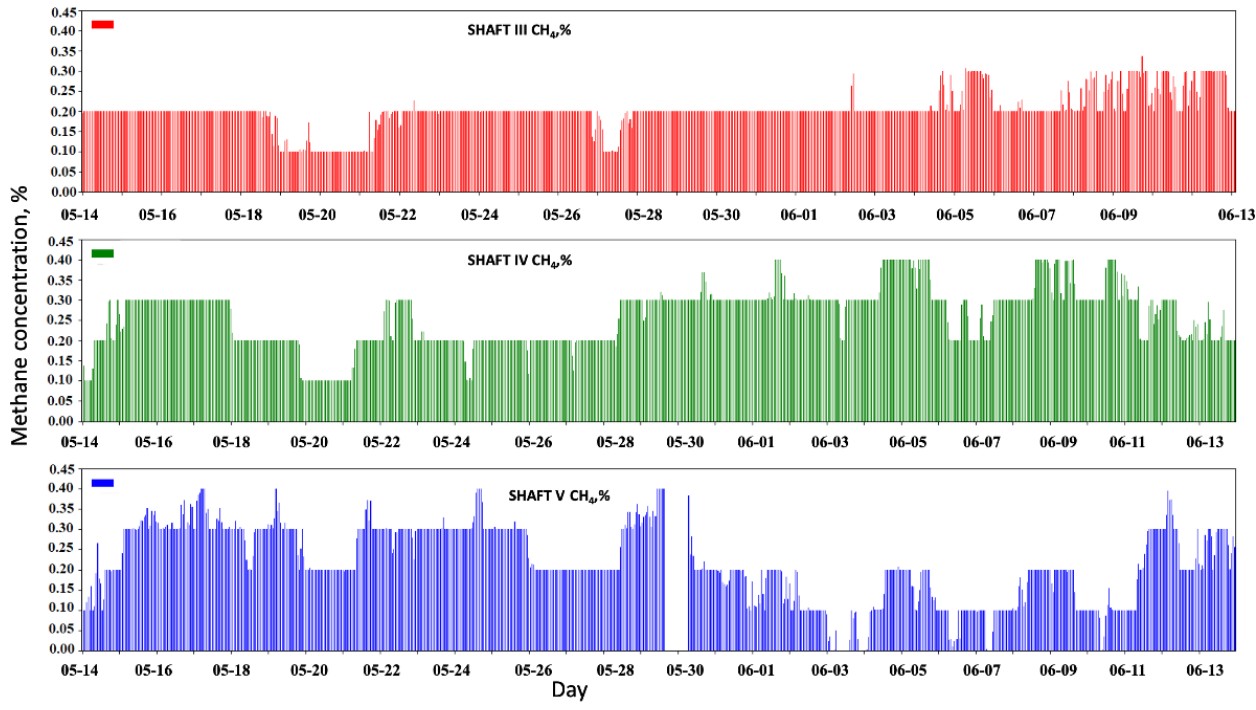

**Figure 7.** Variability of hourly data of methane concentration in Pniówek V and IV coal mine from May 14 to June 13, 2018.

The temporary decreases and increases in the Pniówek shafts are linked to the current extent of mining activities, mainly the value of coal output. According to the ventilation department at that time, the coal mine struggled with the exploitation of the high methane-prone longwall panels. In addition, due to methane exceedances in the underground atmosphere, there were multiple technological breaks and temporary suspensions of mining works. Therefore, we assume that decreased values in Shaft V were caused mainly by reduced mining activities in an individual section of the coal seam excavated at the time. In that case, the permeability of the rock-mass decreased, resulting in a reduced inflow of methane into the underground air, which reflected the gas concentration level in the return air. Similarly, in shaft IV between May 18 and 28, a decrease in concentrations occurred, related to a temporary downtime of mining works and a reduction of output. After that, another increase in methane concentrations was caused by preparing a new section of the coal deposit for exploration. That resulted in the decompression of the rock-mass and releasing more methane after a temporary drop in concentrations.

In addition to methane desorption, methane release can occur due to its outflow from fractures and cracks in the rock-mass caused by mining activity. Such situations are exemplified by instantaneous peaks in methane concentration in the ventilation shaft of Knurów Bojków V, presented in Figure 8. Despite the relatively low concentrations throughout the whole period (0.01 to 0.04 %), there were two noticeable increases on June 2 and 13 (see also Table 2). In the first case, the high concentration of 0.30 % ± 0.1 % lasted for two hours and then dropped to 0.03 %. In the second case, it reached 0.30 % ± 0.1 % and remained at this level for over five hours. Then it dropped to 0.03 %. This situation resulted from an abrupt methane outflow from fractures and cracks in the excavated seam that was ventilated via this shaft.

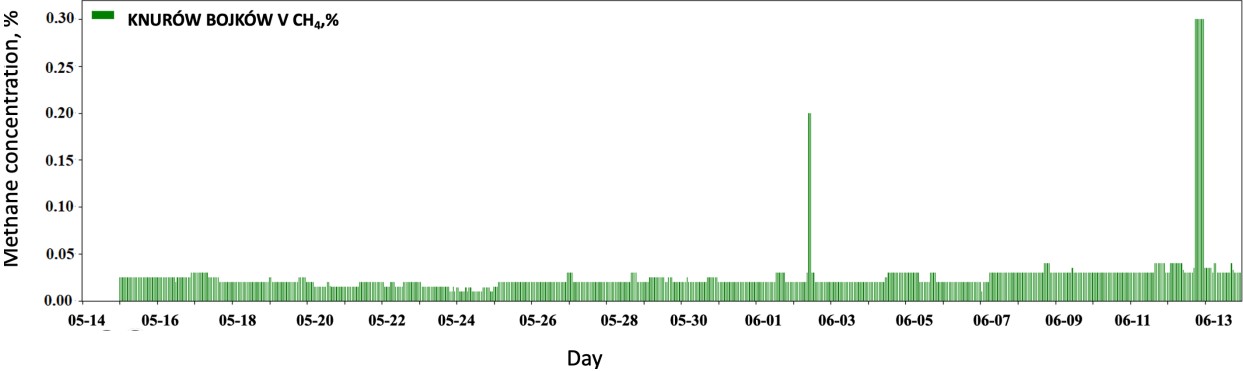

**Figure 8.** Variability of hourly data of methane concentration in the Knurów Bojków V coal mine in the period from May 14 to June 13, 2018

Of all the shafts under analysis, by far the highest values of methane concentration were reported for Budryk V (see Table 2). They ranged from 0.20 to 0.55 % and were subject to considerable fluctuation ($\sigma = 0.11$ %). The gathered data suggests a high methane content of the extracted longwall ventilated by the shaft in question. Although within required limits at all times, such high concentrations suggest that the airflow rate ($\sim$11300 m$^3$ min$^{-1}$; see Fig. 6b) and methane drainage method during the studied period might not have been selected optimally.

## 6        Inventory and temporal data comparison

Commonly available databases of methane emissions from coal mines are usually compiled for individual facilities, without division into their shafts or methane drainage stations. For the sake of comparison with WUG and E-PRTR inventories, in this chapter the emissions determined on the basis of temporal data from the monitoring system for each shaft were summed up in accordance with the specific coal mine they come from (see Fig. 9 and Table 4).

The presented temporal data covering one month reveal that in total coal mines discharged between 186.82 to 349.40 kg min$^{-1}$ of methane (based on Fig. 9). The highest emission was recorded for Pniówek (Fig. 9, dark blue) and Budryk (Fig. 9, light blue) facilities. Thus, during a month of observation, the analysed coal mines released on average of 390.92 tonnes ($\sigma = 51.03$ t) of methane per day. Assuming constant monthly emissions from each ventilation shaft, the yearly average emission is 142.68 kt yr$^{-1}$ ($\sigma = 18.63$ kt yr$^{-1}$). This figure is 27 % lower than the 197.82 kt yr$^{-1}$ reported by WUG inventory (WUG, 2019). In the case of E-PRTR, the difference is even more significant and reaches 36 % (CoMet E.D. v4, 2021), with higher E-PRTR emissions. The main reason is that temporal data presented in this paper includes emissions only from ventilation shafts, excluding methane released from the drainage station (not utilised). It is, however, included in the E-PRTR inventory.

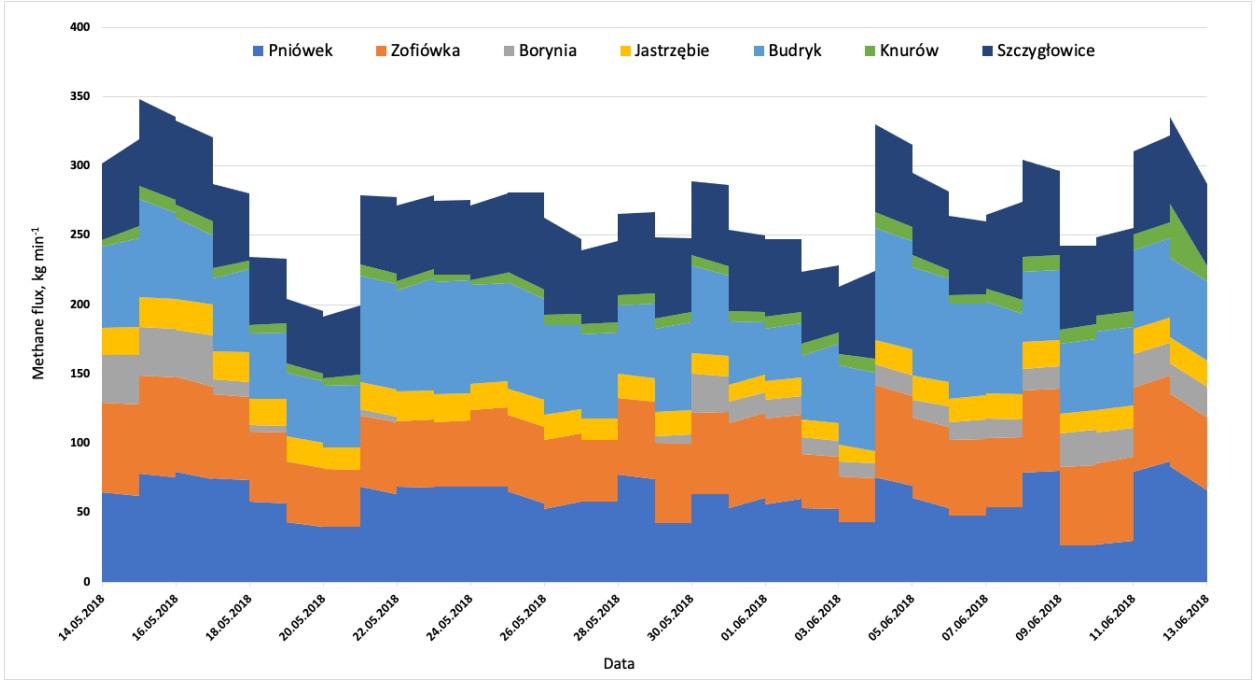

**Figure 9.** Summary of the methane amount discharged from individual coal mines based on temporal data

As was mentioned before, disadvantages of the bottom-up inventories, such as E-PRTR and WUG, are that data reported there refers to entire coal mines and not to individual ventilation shafts or methane drainage stations, and that it is annual data. Furthermore, in the WUG register, annual methane emissions are reported on an even higher administrative level, which means that for the combined entities, such as Zofiówka-Borynia-Jastrzębie and Knurów-Szczygłowice, only one value is given for each entity (see Table 4), without detailed information on the distribution

of the emissions between these fronts. For this reason, the temporal data from these entities has been summed up (see Table 4, the second column in the temporal data). The discrepancy between temporal data and WUG inventory for those coal mines ranges from 1 % to 5 %. We can then assume that verifying measurement data based on e.g. in-situ aircraft measurements using a mass balance or model-based approach on a regional scale (for the entire USCB region) with inventories yields a quite high level of consistency (Fiehn et al., 2020; Kostinek et al., 2020).

Nevertheless, serious problems with determining point sources of emissions arise on the local scale because this approach might underestimate or overestimate emissions from individual shafts. For example, in the CoMet v4.0 internal dataset for comparative purposes, in the absence of temporal data for individual coal mine facilities the data in the WUG and E-PRTR registers were divided evenly between each ventilation shafts. Significant variations between individual shafts were found when comparing in-situ aircraft measurements using a model-based approach (Kostinek 645 et al., 2020) with E-PRTR. The differences resulted mainly from the equal distribution of methane emissions among individual shafts in the registers. As can be seen from the analyses of instantaneous changes in methane fluxes (Fig. 6c and Table 2) cited in the article, the values vary greatly among the shafts of the same coal mine. Therefore, it is necessary to know their specific values in order to accurately verify emissions from individual shafts, e.g. by means of airborne measurements. For this reason, measurement data should preferably be compared with temporally resolved 650 data from the source of methane emissions.

**Table 4.** Comparison of annual emissions of methane extrapolated using available temporal data, aggregated from individual shaft data to the level of individual coal mines (for comparison against E-PRTR inventory) or combined entities (for comparison against WUG). Differences against reported values are also given.


| | Extrapolated from available hourly data | | E-PRTR | Difference | WUG | Difference |
|---|---|---|---|---|---|---|
| | Ind. Mine | Comb. ent. | | | | |
| | kt yr$^{-1}$ | kt yr$^{-1}$ | kt yr$^{-1}$ | % | kt yr$^{-1}$ | % |
| Pniówek | 32.12 | 32.12 | 54.70 | 41.3 | 49.19 | 34.7 |
| Zofiówka | 28.99 | | 27.80 | -4.3 | | |
| Borynia | 8.93 | 47.1 | 12.80 | 30.2 | 46.42 | -1.5 |
| Jastrzębie | 9.18 | | 8.10 | -13.3 | | |
| Budryk | 31.54 | 31.54 | 78.40 | 59.8 | 66.14 | 52.3 |
| Knurów | 4.45 | 34.3 | 4.86 | 8.4 | 36.05 | 4.9 |
| Szczygłowice | 29.85 | | 37.50 | 20.4 | | |

Table 4 presents discrepancies in analysed temporal data with both WUG and E-PRTR inventories which result from several factors. The most important is the assumption that methane concentrations in the ventilation shafts are 660 stable, which is not valid. They change across a year, a month, and even across a day (see Fig.6a). Assuming that they are stable throughout the year, based on the analysed period, leads to underestimating methane emissions (Budryk and Pniówek), or overestimating them (Jastrzębie or Zofiówka) (see Table 4). A high discrepancy in Borynia is probably related to the malfunction of the methane sensor in shaft VI between May 17 to 30 (according to the source file). Other

reasons for discrepancies are the methodology, frequency, and timing of measurements.

The most common method for methane concentration measurements used in Polish coal mines is combining a handheld anemometer to measure airflow velocity with air samples analysed in the laboratory (see chapter 3). Such measurements are required to meet statutory safety regulations and have to be done by a trained person following recognised procedures. Furthermore, they can be used for methane emission monitoring, provided they are taken during production shifts (UNECE, 2021). Nonetheless, these measurements are conducted only one day per month,

which means that annual methane emissions reported in WUG and E-PRTR are calculated based on twelve measurements only. Therefore, comparing data from inventories with temporal data can lead to differences, like in the case of the Budryk coal mine (see Table 4). The key factor is also the timing of the measurement. Because methane emissions into mining excavations depend mainly on the scope of mining activity (see chapter 5), methane concentration measurements should not be carried out during the shift changes or after the mine has stopped production,

which leads to a decrease in the values.

The preferable way of measuring methane concentrations would be using the methane fire teletransmission monitoring system (SMP/NT), because it allows for continuous methane measurements and eliminates the problem related to the frequency and timing of measurements. More reliable results are obtained by continuously measuring the concentrations of methane and air fluxes, which can then serve to validate aircraft or satellite measurements.

However, the monitoring system mentioned above aims to check that preset safety criteria are met. On the other hand, potential emission monitoring should allow for gas flow quantification with high precision. Therefore, before its implementation, it should be suitable for high-precision methane measurements. It is worth considering the replacement of pellistor sensors with more precise ones. Modern optical instruments (TDLAS), based on the absorption of infrared light at a single wavelength (e.g. 1.62μm), can determine the methane concentration in the air

directly above the ventilation shaft exit with an accuracy of 0.27 % and sensitivity of 25 ppm (Gao et al., 2013).

In addition, let us examine the Pniówek coal mine - the only coal mine that uses a monitoring system to measure methane emissions. Even a methane fire teletransmission monitoring system can underestimate emissions if the frequency of measurements is inadequate. Recording the methane concentrations as an average from one day per month only, considering their high variability (see Fig.6a), does not provide reliable results. A prerequisite for using

this system would be continuous methane concentrations measurements and reporting them monthly as a weighted average, taking as a weight the time of a given concentration in the ventilation shaft.

There is no doubt that temporal emissions data from individual ventilation shafts would provide valuable information for verifying top-down approaches. Nonetheless, all the above-described factors can significantly impact the magnitude of the differences when comparing instantaneous emission results with measurement data obtained with

instruments on board of aircraft. That being said, it is essential that the data for the verification should be reliable and reflect the actual emissions values from individual point sources as much as possible.

**7      Summary and conclusion**

Accurate determination of methane emissions requires an integrated monitoring system primarily based on a combination of top-down and bottom-up approaches. In addition, accurately determining emissions from an individual

ventilation shaft on the local scale requires instantaneous data, which is very difficult to obtain due to the lack of such inventories. As part of this article, we analysed temporal emission data for 15 ventilation shafts of underground coal mines in the USCB area during the CoMet 1.0 mission and determined the factors influencing their variability. The methane concentrations in examined shafts ranged from 0.00 to $0.55 \pm 0.1\%$ and were subject to a significant variation on a day-to-day basis. The main factors that influence the concentrations and emissions variability are saturation of the individual seams with methane (methane content), the permeability of the rock-mass, the scope of mining works performed at the excavated longwalls (coal output), and the abrupt outflow of methane from fractures and cracks.

The presented temporal data for the CoMet 1.0 observation period revealed that the studied individual coal mines released between 186.82 to 349.40 kg min$^{-1}$ of methane, resulting in an average emission of 390.92 tonnes ($\sigma$= 51.03 tonnes) per day for all investigated mines. Conversion of this number for 12 months provided the average emission at a level of 142.68 kt yr$^{-1}$ ($\sigma$= 18.63 kt yr$^{-1}$), which is lower than the WUG data by 27% and the E-PRTR data by 36%, respectively. Additionally, data for individual coal mines were both over- and underestimated. The discrepancies between temporal data and both analysed inventories result from the assumption that the methane concentrations in the ventilation shafts are stable, which is not valid, as well as from the methodology, frequency and the timing of measurements. All the above-described factors can significantly impact the magnitude of the differences when comparing instantaneous emission results with measurement data obtained with instruments on board of aircraft. That being said, it is essential that the data used in top-down emission estimation studies should be reliable and reflect the actual emissions values from individual point sources as much as possible. Such data should be available at relevant time scales (down to hourly), and the methane emissions should be estimated with the greatest possible accuracy.

It is possible to achieve this by using a standardised emissions measurement system used for safety purposes at all coal mines. In case of Poland the source of data would be SMP-NT/A methane monitoring system which all coal mines are equipped with. Although the system is aimed at monitoring the preset legal criteria of methane concentrations, it could be easily customised for gas flow quantification. Increased precision could be obtained by replacing the existing methane sensors with more precise types, although that would require extra investments that will not be considered cost-effective, thus would have to be required by law. Flow monitoring networks would also need to be expanded.

More analyses would be required before high-frequency temporal data from the safety systems can replace the well-established methodologies used for annual reporting. In this context, dedicated measurements of emissions from individual ventilation shafts and methane drainage stations would be highly beneficial for validation purposes. It should also be noted, that before these new data streams are directly comparable to reported values, emissions of non-utilised methane from the methane drainage stations need to be considered. This could provide additional challenges, as these can intermittent rather than continuous.

*Data availability*. Emission data presented here form one of the core components of the data set "Emissions of CH4 and CO2 over the Upper Silesian Coal Basin (Poland) and its vicinity (4.01)" available at https://doi.org/10.18160/3k6z-4h73.

*Author contributions*: JS developed a methodology for the presentation of research results, contributed analysis tools, analyzed data and wrote the paper (60%), AF developed a concept for the presentation of research results and participated with manuscript editing (15%), MG contributed analysis tools, co-created CoMet 4.0 dataset and edited manuscript (25%)

*Special issue statement.* This article is part of the special issue "CoMet: a mission to improve our understanding and to better quantify the carbon dioxide and methane cycles." It is not associated with a conference.

*Acknowledgements*:

*Financial support*: MG has been supported by the Max Planck Society (MPG), and the German Aerospace Center (DLR) . MG and AF acknowledge financial support from German Federal Ministry of Education and Research (BMBF) through project AIRSPACE (grants no. FKZ 390 01LK1701A and FKZ 390 01LK1701C)

CoMet 1.0 received support from German Science Foundation (Deutsche Forschungsgemeinschaft, DFG) within DFG Priority Program SP 1294 "Atmospheric and Earth System Research with the Research Aircraft HALO (High Altitude and Long Range Research Aircraft)".

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
