# Peer review of "Factors influencing the temporal variability of atmospheric methane emissions from Upper Silesia coal mines: A case study from CoMet mission."

_Atmospheric Chemistry and Physics, 2022_

## Author Comment (AC1)

**Authors' response to the comments from Reviewer #1**

(Title: 'Review of "Factors that influence the temporal variability of atmospheric methane emissions from Upper Silesia coal mines: A case study from CoMet mission", Swolkien et al., for ACP.')

**Authors' general comment:**

*We would like to express our thanks to the reviewer. We hope that the changes in the second version made it more straightforward for a reader and expect that the manuscript will now become acceptable for publication in the Journal.*

*Here we address all the comments from the reviewer. In the following, comments from the reviewer will be marked with* regular font, *while our responses are given using italic font.*

*For clarity of structure, we have divided the discussion into three sections: major comments, specific comments and technical corrections.*

**Reviewer 1, Major Comments**

*We have divided the major comment from the reviewer into two separate issues.*

**Issue 1**: This paper presents temporally-resolved (mostly hourly) measurements of methane flux from 15 coal mine ventilation shafts in Poland, pointing out that temporal variability might be negatively impacting the inventory calculations which are based on much less frequent (unclear how much less) measurements or accounting. Overall, the measurements should be published, but I only recommend this manuscript after significant revisions for clarity mostly, but also I think the authors need to make their final points in a better way so they have more impact on the reader.

One of my main comments is that someone should carefully review the text for English language & clarity. Some sentences are awkward, and although I have noted some of them specifically below, I have not done so for all the language/grammar issues. Often (for example, the first paragraph of the summary), the poor English impacts the readability of the text and makes it impossible to understand the point. After that, there were a few places where it was not really clear where they are getting some of their numbers for final emissions (I've noted those below).

*The authors applied all the grammar and spelling corrections suggested by the reviewer in their review. Additionally, following the reviewers' advice, the authors rewrote the sections that were pointed out as not clear enough. The authors also tried to be more precise in the statements that were presented in the paper. After all corrections, the paper was handed over to be re-checked by a native speaker for stylistic and grammatical errors. We have addressed the specific comments in the dedicated sections below.*

**Issue 2**: The paper is organized fairly well but the discussion/conclusions should be strengthened to make some broader points more clearly. It's not obvious why top-down measurements (for example for an aircraft-based mass balance approach) would be incorrect exactly, or if it is correct.

*As shown in the studies by Fiehn et al. (2020) and Kostinek et al. (2020), the results obtained using aircraft-based mass balance method give relatively accurate (when compared to the bottom-up reported*

*values) results on a regional scale, i.e. for the flux over the entire USCB (Upper Silesian Coal Basin) area (both studies were carried out based on results obtained during CoMet 1.0 campaign in 2018). In the first study, the authors showed that CH₄ emissions estimates from two flights were in the lower range of the six presented emission inventories (Fiehn et al., 2020). In the second case, derived emission rates coincided (±2 %) with annual-average inventorial data from E-PRTR 2017, but they were distinctly lower (-37 % / -40 %) than values reported in EDGAR v4.3.2 (Kostinek et al., 2020).*

*However, a larger issue in comparing the results of fluxes derived using the measurement-driven top-down approach arises when attempting to quantify emissions from a single source, such as an individual coal mine ventilation shaft. Thus, in the studies where only one source is (or a small number of sources are) studied in the same method, using annual databases for this purpose may lead to overestimating or underestimating the methane fluxes.*

*As is shown in the article, the methane fluxes from individual shafts can be highly variable. While this variability can have serious consequences for the top-down flux estimation studies using a temporally-limited observational data, so far it was rarely been taken into the account. To the best of our knowledge, this is the first study that provides and discusses hourly emission data for the purpose of comparison against independent emission validation methods.*

*Data presented in our paper provide insight into the magnitude of the variability over a well-studied area of USCB, that is an important step towards more accurate emission estimations on both local (individual shaft) and regional (study area) scales. We hope that it will also be extremely useful for future studies performed over areas where such highly-resolved emission data cannot be obtained.*

*We have commented on this in the revised text, e.g. in lines L119-127 and lines L500-512.*

**Reviewer 1, Specific Comments**

**Issue 3**: L34: Should this read "assumption that methane *emissions* are time-invariant", rather than "concentrations"? I would think most methods for estimating emissions using concentration measurements still account for wind speed (if atmospheric) or air flow rate for example.

*Indeed, the wind speed is a critical factor in estimating the methane emissions using measurements in the atmosphere (e.g. using airborne measurements).*

*However, in L34, and also in L109-110, the concentration of methane inside the mine ventilation shaft was meant. As the relation between these concentrations and emissions to the atmosphere are usually co-dependent, we agree that "emissions" would be acceptable here. For more details, please see discussion in 'Issue 5' below.*

**Issue 4**: L80-81: isn't FTIR an example of remote sensing, already mentioned? Perhaps rephrase to mention satellite vs. ground-based remote sensing specifically?

*This section describing measurement observation has been rewritten and can be found in L79-85 in the revised manuscript.*

**Issue 5:** (*Linked following two comments*):

L109, grammar - whose importance? (what does "their" refer to? The factors that influence variability (concentration - should this be emissions?
L110 - presumably concentrations in the shaft are directly related to emissions? (is that true?).

*Yes, indeed. The variation of concentrations in the ventilation shaft causes variation in the emissions to the atmosphere.*

*The variability of methane concentrations in ventilation shafts (measured 10 to 15 m below the outlet of the shaft) results from mining-related factors, i.e. the nature and scope of the mining works performed in the underground section that is ventilated by a particular shaft. Naturally, any increase in airflow will dilute the methane to its lower concentration, but in reality the airflow variations are usually small and not as significant as that of methane concentrations. Thus, the emissions are most of the time (but not always!) correlated to methane concentrations in outflow air passing through the shaft.*

*Generally speaking, airflows in operational can be very stable over days or sometimes even weeks (Fig. 6b in the revised manuscript). More critical from the source-estimation perspective are situations where production is significantly limited due to an emergency, e.g. in case of an accident related to a sudden methane outflow. In such scenario, mine operators can significantly alter the mine's ventilation network functionality, and in effect – methane emissions from a given mine shaft. However, regular operations can also cause significant variability of emissions, as can be seen in Fig 7 of the revised manuscript.*

*However, while we analyse the concentration data in the paper, that usual correlation can be assumed and discussing emissions is better for readability in this paragraph. It has been revised in the manuscript, see L119-127.*

**Issue 6:** L138. Is 656 the WUG estimate supposed to correspond to the 604 kt from UNFCCC? (i.e. the estimates are of the same thing but estimated slightly differently using different methods?).

*Unfortunately, the datasets mentioned above are not directly comparable. The value determined by the WUG corresponds to the methane emission from the rock mass, i.e. it compares to the so-called CMM total methane emission (see lines 163-166 for an explanation). The value of 656 kt corresponds to the amount of methane captured by methane drainage and released directly into the atmosphere (VAM). The value of 604 kt (UNFCCC- atmospheric methane emissions) corresponds to the total methane released into the atmosphere. Still, it also includes the amount of methane, calculated based on emission factors (taken by UNFCCC), emitted during post-mining activities and from the abounded coal mines.*

*In the WUG report, atmospheric methane emission is understood as a sum of VAM and non-utilised methane and equals for 2018 511 kt of $CH_4$ (see Table 1). However, an important caveat is that this number does not include methane captured by the drainage system and subsequently utilised. It only consists of this portion of methane that was not utilised and directly released into the atmosphere. This happens often, mainly due to lack of storage capacity of the drainage installations but is highly variable in time and difficult to quantify. Nevertheless, emissions from these sources are estimated and included in the reports for UNFCCC, thus also preventing comparison of VAM from WUG reports to UNFCCC.*

*The authors added the explanation in line L159-163 in brackets.*

**Issue 7:** L157. this was confusing until I realized that the combined entities are referred to as fronts, not the individual facilities - the parentheses are vague as to which they refer to. Actually re-reading I am still not sure what this means.

*We apologize for any confusion. The administrative attribution of coal-mines for reporting in Poland is quite complicated. Generally, mines can function in two ways:*
*- they can operate as individual facilities, such as the Pniówek or Budryk mine, or*
*- they are administratively combined into one larger enterprise (combined entities), such as Knurów-Szczygłowice or Borynia-Zofiówka-Jastrzębie.*

*In the second case, the enterprise level was created for purely administrative reasons. In practice, each of these mines function in the same manner as any individual one (like Pniówek or Budryk mentioned above), but inside the combined entity it is then called a 'front', as the 'mine' term is reserved for the larger enterprise. For example, the Borynia-Zofiówka-Jastrzębie combined mine consists of three fronts: the Borynia Front, the Zofiówka Front and the Jastrzębie Front. At some point in the past each of those fronts were individual mines, with separate entries in the emission reports.*

*Regarding the E-PRTR reporting scheme, this administrative reshuffling should not (theoretically) matter, as each front should report emissions individually, however inconsistencies in reporting across the years were observed. To make matters even more complicated, in the case of the WUG database the reporting is required from the combined-entities level (see e.g. Table 4 in the manuscript). In other words, in the WUG reports for the Borynia-Zofiówka-Jastrzębnie mine, there is only one emission value.*

*The fragment of the manuscript concerning these details has been rewritten – see L175-182.*

**Issue 8:** L335: Why is the methane emission not calculated for each return and then summed, rather than summing the airflow and averaging all the concentration measurements in a weighted fashion? As long as they are weighted by the airflow (presumably that is what is meant here), this is the same procedure (result) but much easier to explain.

Also, why could these measurements not be done in the ventilation shaft itself so that fewer measurements would be needed?

*Indeed, the procedure suggested by the reviewer should be correct and the airflow flowing into the ventilation shaft should theoretically be the sum of the streams from all connecting returns. However, in practice the procedure of estimating annual emissions (described in appropriate law) requires that the averaged concentrations need to be calculated first and only then scaled per flow measured in the ventilation shaft – as described in the manuscript. We assume, that the main reason for that is to correct any potential errors stemming from extra flows that could be present in the mine, however we have not been able to confirm this despite our efforts. The paragraph in question has been corrected to make it more transparent – L349-363 in the revised manuscript.*

*Answering the reviewer's second question: The measured procedure is defined by law. Similar strategies apply in most mines around the world. Measurements of methane concentrations in shafts for reporting purposes require much greater precision and are performed one day a month, and the samples are submitted for laboratory tests. These measurements cannot be performed in a ventilation shaft due to difficult operating conditions - mainly due to high air velocities. In some shafts, the airflow can reach 22,000 $m^3$ / min. Therefore, measurements of concentrations in shafts are performed only sporadically to control and validate measurements done underground, and only the airflow is determined in the ventilation shaft channel.*

**Issue 9:** L340 CH4 is in percent by volume, please note (not by mass). Concentration often refers to a mass per unit volume, so should be clarified here.

*It has been clarified – L361 in revised version.*

**Issue 10:** L351 And where are these measurements made, are they also in the returns or the ventilation shafts? [I see later much more discussion of this, but perhaps earlier on this could be mentioned. How does the methane sensor "protect" faces & longwalls?] Maybe make this a little clearer.

*We've clarified the text in L364-372*

**Issue 11:** ~L400: Can the authors compare the approximate accuracy of this measurement with that of the other method that occurs quarterly?

*It is very difficult to estimate and compare these two measurement uncertainties.*

*Unfortunately, annual emission values are provided in the publicly available databases without uncertainty. Similarly, mines also generally do not estimate the uncertainty of emissions as they are not required to do so. Several factors may contribute to the overall emission uncertainty:*

> *i)      temporal variability of emissions, not captured by monthly measurement regime.*
> *ii)     relatively high potential for 'human error' with manual sampling procedures,*
> *iii)    extra inflows of methane into the collection shaft not captured by manual methods.*

*Using widely available safety systems may limit the impact of those uncertainties on the accuracy of the emission estimates, especially with regard to i) and ii).*

*Thus, precise estimation of the uncertainty would require a dedicated study based on a longer dataset, which is not available and thus outside of scope of our current study.*

*Also, we would like to point out that in all the Polish coal mines, the precise emission estimation occurs on a monthly, not quarterly basis.*

**Issue 12:** Also, when the uncertainty is stated as 0.1%, that is 0.1% of CH4 concentration, not as a percentage of the value? (as later it is stated the uncertainties are close to 20% on the flux).

*The value of 0.1% is the absolute instrument's uncertainty, as specified by its manufacturer. Of course, it concerns the concentration of methane, not a percentage of the value, e.g. the measured concentration will be, e.g. 0.2% +/- 0.1%. In such case the relative uncertainty would be 50 %, but usually the relative uncertainties are lower. We have clarified this in L395-397.*

*The uncertainty reported later (20%) is relative. It results from the harsh conditions in which methane sensors operate. For example, the air humidity in ventilation shafts is very high, the sensors are prone to flooding, and there are very high air velocities. All these factors significantly contribute to the final value of methane emissions calculated based on concentration.*

**Issue 13:** L469-470 I'm not sure I follow the logic of why the fact that the peak concentration is higher than the average is important for verifying aircraft data - perhaps another sentence to connect these dots is needed?

*Strong fluctuation of concentration (hour to hour) can influence the estimated emissions. Emissions peaks that last several hours can easily be detected by instrument aboard the aircraft, especially if the measurements are performed in relative vicinity of the sources (the definition of 'vicinity' here depends on the atmospheric conditions, but could be measured in hundreds of kilometers under some conditions). The data measured by airborne (but also other platforms) usually covers only several hours of observations, thus rapid changes of emissions can cause discrepancies.*

*Until now, verification of methane emissions has always assumed that emissions from coal mines were constant and equal to the annual mean, so that momentary changes do not significantly affect the results. CoMet 1.0 campaign provides an excellent opportunity to test those assumptions and gain insight into the effects of actual emission variability on the accuracy of measurement-driven estimates.*

*The text in lines L505-514 has been expanded and improved.*

**Issue 14:** *(two comments merged)***:**

Figure 8 (a) is not labeled (a).

Figures 7 and 8a & 8b should probably be re-made into a single figure with 3 panels that would go in better order: concentration, air flux, and then methane flux. Also why bars instead of box plots for the air flow figure? I like the box plots for all three if possible; that would allow for understanding the fluctuations in air flow as well as concentration.

*The figures were re-made according to the reviewers' proposal.*

**Issue 15** Table 3 why do some shafts only have monthly averages? Is hourly data not available? How is there a standard deviation calculated on the monthly average when the period in question is mid-may to mid-June, i.e. one month. Is that the standard deviation of two values (one for May and one for June)? Table 3 caption should probably mention the period again.

*Table 3 presents the statistical description of methane fluxes determined based on hourly concentrations (Table 2). The last column in Table 3 relates to the frequency of airflow measurements. In four cases, the average monthly flows were provided directly by the mine employees, but these are still based on safety – system, high frequency data (not monthly measurements for reporting purposes). We have included this data in study for comparison purposes. In the remaining cases, hourly or daily data was shared, depending on availability.*

*As mentioned before, the air flux in individual ventilation shafts is determined by the number of mining regions ventilated with a given shaft, their projected total methane emission, and the scope of mining works performed. An appropriate air flux is determined for each operational longwall.*

*We have clarified the Table 3 caption.*

**Issue 16:** L520 is the uncertainty on these numbers ~0.1% (from earlier?). I.e., when it ranges from 0.1% to 0.2%, that means 0.1%+/-0.1% to 0.2% +/- 0.1%? It seems like a very high uncertainty when looking at these figures - is that correct, or perhaps the authors note if some of this variability is just uncertainty?

*Methane concentration measurement readout precision is only 0.1% vol (i.e. when the sensor reads 0.1%, the measured concentration can be between 0.05% vol or 0.15% vol). It is hard to talk about high precision here, as these sensors are mainly used to control whether the concentration does not exceed the critical value of 0.75% vol (safety margin). In this case, we wanted to show that in a short time, the concentrations may change, for example, from 0.2% to 0.4% vol to, e.g. 0.1 to 0.2% vol due to scope of mining works. Such changes in concentration are best observed in specific mining longwalls, e.g. during periods of production downtime (e.g. during weekends), when the concentrations drop can be observed. After the normal production cycle is restored, the concentrations increase again. These changes are then transferred to the concentration readings in the ventilation shafts, and thus – to atmospheric emissions.*

*See also our comment in Issue 12.*

**Issue 17:** L556 and caption fig. 11, again, this is a language clarity issue - the total of all the mines was between 186 and 349 kg/min, not the emissions from the individual mines. The way this is written, I would have thought that each mine was emitting that much, until I look at the figure.

*Yes, it was a language clarity issue. It was corrected in the text - see L620 in the revised manuscript.*

**Issue 18:** L558 - I do not see 61 or 60 kg/min as a maximum anywhere in table 3?

*Wording issue. These values correspond to the sum of CH4 flux averages from these two mines (from three shafts of Pniówek and two shafts of Borynia); they have not been listed separately. A minor mistake in calculation caused the sum from the table to not be equal to the numbers provided. The authors apologize for this mistake.*

*For clarity, we have removed the numbers from the updated manuscript and simply stated that two mines discharged the most methane in the studied period – see L619.*

**Issue 19:** L560, how is this number arrived at? (142.68 kt/yr?). Is the average over all the mines 390.92 kt/day, how is that equal to 142.68 kt/year? what am I missing here? (a unit issue most likely?).

*Apologies, indeed a mistake with the unit. it should be:390 t CH4 per day. It was corrected in L621.*

**Issue 20:** Table 4 - seems to be an extra column for some of the shafts. Not clear what this is, perhaps explain in the title.

*Updated the label and the table itself for enhanced clarity. The explanation was also added in the text – L633-635*

*A more in-depth explanation of the reasons on different reporting strategies is given in the response to Issue 7.*

**Issue 21:** L587 - is the more important assumption rather that the emissions from the shafts are stable? (i.e. we know the air flow rate is changing as well, so the flux is what is important here, right?). fig 11 shows fluxes, not concentrations. This is a question actually, some of the text indicates airflow varies between mines (L486), but perhaps is more constant in time than the concentration. This could just be stated more clearly when the authors use concentration to stand in for flux. I would think that if a mine has an event with a high emission, the airflow is increased in order to maintain safe concentration levels so there would be some variability there.

*See our previous explanations (e.g. in issues 3, 5 and 8). Airflow in shafts varies between mines because each coal mine has different numbers of operated longwalls, so the need for air is also different. Nonetheless, air flux in particular shafts does not change a lot monthly. The authors added an explanation in lines L521-525 and 545-547*

**Issue 22:** L587+ Does either inventory method use the quarterly measurements in the ventilation shafts described earlier and also described here in line 593? L589 suggests that the inventories (MUG and EPRTR) are using measurements from "only one month" to estimate across the entire year, but later L599 it is indicated they are based on a single measurement for each month. Please clarify.

*The authors added an explanation in lines L669 -670 and 686-691 The analysis conducted in the article on instantaneous data from the monitoring system covered one month. Therefore, measurements should be made over a more extended period. On the other hand, the traditional method measures the methane concentration only one day per month (note: not per quarter), which means that the yearly data reported to the WUG or E-PRTR inventories are calculated based on 12 days of measurements (please note that in the studied mines these are monthly, not quarterly as the reviewer assumes). Increasing the frequency is a logistical challenge, as it would require the employment of additional staff, which would be challenging. For this reason, the authors suggest using a monitoring system that provides continuous concentration measurement without hiring additional employees. Of course, prior to that, it would require being adapted to more precise measurements, e.g. by replacing the sensors with more precise ones, as discussed in the text.*

**Issue 23:** L603 - ? unclear grammar but is this saying that the hourly fire telemetry data is not as accurate as the monthly or quarterly data? So would using the less accurate hourly data still be an improvement over the less frequent but more accurate existing measurements? This is an important question if this is what is being advocated, or is the paper advocating for improving the fire telemetry data (at what cost?) in order to get better averages? Also, if the telemetry data is not accurate enough for reporting, does that undermine the entire study here, because we cannot believe the hourly data over the current quarterly measurements?

*See previous also comments to previous issues.*

*We agree that the less accurate data from the safety system (note: not always available at hourly intervals) is not immediately applicable for annual emission reporting. This study should be treated only as an initial step towards using such hourly data. Authors' primary concern was the potential for over- and underestimations when validating annual emissions using time-limited atmospheric observations collected during CoMet 1.0, subsequently used in the framework of top-down estimation methods.*

*For this purpose, in the authors opinion, continuous monitoring system is better than the average of 12 measurements throughout the entire year directed to the WUG or E-PRTR inventories, as the latter do not take into the account variability of emissions at time scales relevant for limited-time campaigns.*

*Of course, the costs of adapting the system to more precise measurements, e.g. by replacing sensors, should be considered.*

*We have clarified this in the revised manuscript (e.g. L720-731)*

**Issue 24:** L631 Where are these 61 and 60 kg/min numbers from, they are also in L558 presented as the maximum values for 2 specific shafts, and here they are presented as a range of values from all the mines. 60 kg/min * 60 minutes/hour * 24 hrs/day = 86,400 kg/day = 86 tons per day. The annual number of 142 kt/yr also does not match any of these, although it does work if the number is 390.92 tons (not kt) per day. This math needs to be clarified, units checked, etc. here and in the rest of the paper!

*The discrepancy stemmed from wrong units (should be 390.92 t per day). Detailed explanations regarding the numbers mentioned above are provided in **Issue 19**.*

*Of course, the authors agree with the comments of the reviewer. All numbers presented in the article were re-verified and corrected if necessary.*

**Reviewer 1, Technical corrections**

L48 awkward/grammar "when additional carbon footprint is considered". - *Corrected.*

L56: emission should be emissions - *Corrected.*

L68: omit "the" before "use" - *Done.*

L83-84: awkward grammar again. - *Corrected.*

L84: try "obtaining temporally-resolved emissions from " - *Sentence rewritten.*

L91: again it seems that this should say "emissions" rather than "concentrations"? - *See comment above (Issue 5). Sentence rewritten.*

L98-99+: grammar issues with this list. - *Corrected.*

L102: the goal was - *Corrected.*

L133 – 41% of total European emissions (add European for clarity here) - *Corrected.*

L136 add "the" before "European" - *Corrected.*

L139 insert "the" before "country's" - *Corrected.*

L143 what does "these" refer to here (resources or seams?) - *This sentence has been rewritten – L167-168.*

Fig 3b the legend in the figure indicates that the boxes are the 95% CI, but the caption says it's the 25% to 75% percentile, which is more standard for a box plot. - *Corrected, the legend now reads 25% - 75%.*

L340 shouldn't $Q_{air}$ be in m^3 air per minute, not $CH_4$? (or change to $Q_{ch4}$). - *Corrected.*

L346 "uses it to report it" should be reworded*. - Done.*

Fig 5 pressure is misspelled - *Corrected.*

L435 spelling appropriate - *Corrected.*

L436 "analyzes" - > analysis? - *Agreed.*

L496 much more complex than what? - *Corrected.*

L511. awkward last sentence, perhaps rephrase: "Increased mining activities will always result in methane outflow, regardless of the form the outflow takes"? - *Corrected.*

L631 revolved is not the right word here? Maybe revealed? - *Corrected*

L634. Both over *and* underestimated - *Corrected.*

L645 - also air flow rates would need to be part of this yes? - *Added in L724.*

---

## Author Comment (AC2)

**Authors' response to the comments from Reviewer #2**

(Title: 'Referee comment on "Factors that influence the temporal variability of atmospheric methane emissions from Upper Silesia coal mines: A case study from CoMet mission" by Justyna Swolkień et al., Atmos. Chem. Phys. Discuss., https://doi.org/10.5194/acp-2022-243-RC2, 2022)

**Authors' general comment:**

*We would like to express our thanks for the review of our paper. We hope that the authors' changes in the second version made it more straightforward for a reader and expect that the manuscript will now become acceptable for publication in the Journal.*

*The authors applied all the grammar and spelling corrections suggested by the reviewer in the present review – the minor corrections are not listed below for clarity. Also, following the reviewers' advice, the authors rewrote the sections that were pointed out as not clear enough. The authors also tried to be more precise in the statements that were presented in the paper. After all corrections, the paper was handed over to be re-checked by a native speaker for stylistic and grammatical errors.*

*Now we would like to refer to the more specific comments from the reviewer. In the following, comments from the reviewer will be marked with* regular font*, and our responses are written in* italic.

**Reviewer 2, Specific Comments**

**Issue 1**: In the introduction the explanation for the motivation for this work could be improved. Are there discrepancies between the inventories and top down studies for this source? Is this a demonstration of a method for emissions quantification that could be widely used in other mines?

*One of the potential major issues in comparing the results of reported (bottom-up) emissions with measurements-based (top-down) estimations arises when attempting to quantify emissions from single sources, such as coal-mine ventilation shafts. The use of annual databases for this purpose may lead to overestimating or underestimating individual sources' share, due to the potential for significant temporal changes in methane fluxes, as demonstrated in the manuscript. These changes can often be diverse one coal mine, as ventilation shafts that emit methane can be turned on and off over the period of activity for various reasons related to the day-to-day mining operations. To accurately verify annual emissions from individual mines with observations limited temporally (as is the case when performing airborne measurement campaigns like CoMet 1.0), it is necessary to know whether such temporal changes have occurred. Additionally, having access to temporally resolved emissions provides a unique opportunity to validate the accuracy and precision of various flux estimation techniques, including mass-balance methods and Bayesian inverse-modelling.*

*It should be noted that the results obtained using aforementioned techniques can still give relatively good results when the regional scale is considered, i.e. the entire USCB area. Such a comparison was made by Fiehn et al., 2020 and Kostinek et al., 2020 for the entire USCB during CoMet 1.0. In the first case, the authors showed that $CH_4$ emissions estimates from two flights were in the lower range of the six presented emission inventories (Fiehn et al., 2020). In the second case, derived emission rates*

*coincided (±2 %) with annual-average inventorial data from E-PRTR 2017, but they were distinctly lower (-37 % / -40 %) than values reported in EDGAR v4.3.2 (Kostinek et al., 2020).*

*However, in order to increase the estimation accuracy when using observational-based methods, more precise data should be obtained. The authors are convinced that using the safety parameters monitoring system for providing continuous information on methane emissions might be a relatively 'cheap' solution. Currently, not only all mines in Poland, but also across the world, are equipped with such systems.*

*Unfortunately, it is not possible at the moment to provide a detailed comparison between the monthly emissions reported to the national inventories and hourly data from the monitoring system. Therefore, the authors have limited the scope of the study to demonstration of how these hourly emissions varied during a limited (month) time of the CoMet campaign. At the moment, this can only be treated as a first step towards using safety-monitoring systems as means of providing highly-resolved (in time) methane emissions, as these would potentially require calibrations and statistical analyses over a longer period, and in some cases sensor replacements (a major investments for some enterprises). We have mentioned these issues in L708-719 of the revised manuscript.*

*It should also be noted that at the moment, high-frequency data needed to estimate hourly emissions from monitoring systems are not easily obtainable, as the companies operating the mines are not required to provide these datasets. The data presented in our study have been made available upon request from a limited number of mines (albeit important regional emitters).*

*Nevertheless, the authors believe that the continuous monitoring system can provide important advantages over the law-sanctioned estimates based on average of 12 measurements throughout the year (as directed in Poland by the protocols of WUG or E-PRTR reporting).*

*We have attempted to clarify this reasoning throughout the manuscript, e.g. in L119-127, L500-512, L708-719.*

**Issue 2**: I think a little more information could be given on the sensors used to make the methane measurements. Many readers will be unfamiliar with the types of sensor used in the SMP-NT/A monitoring systems. Are the large fluctuations in concentrations at individual sites real, or related to measurement precision? Is there any potential to use other higher precision sensors for methane concentration measurements? Would there be much gain in doing so?

*The authors added additional information about the sensor in lines L428-437. The methane sensors described in the paper are part of the SMP-NT/A monitoring system and are used in mines as devices to control whether methane concentrations do not exceed the legal safety limit of 0.75% (vol.). These measurements are usually characterised by high uncertainty, translating to higher uncertainty of methane emissions. Due to that, they were not historically used for reporting emissions.*

*Theoretically, it is possible to use more precise sensors, e.g. TDLAS (tunable laser diode absorption spectrometer, open path or closed path\*) analyser, directly over the ventilation shaft diffuser. Usage of such systems can theoretically allow to provide detailed information on fluxes on temporal scales of seconds. However, due to lack of requirements and high costs, these have never been used in any coal mines, and at the moment it is not planned to require it. It is also difficult to determine whether these instruments would be able to operate correctly in the supersaturation conditions and with a very high air flux (sometimes even 23000 $m^3$/min) of the upper parts of the ventilation shafts.*

Methodology of precise estimation of emission from coal mines are currently investigated. The possibility of using the sensors mentioned above in the ventilation shafts of selected mines in USCB is the subject of a preliminary research project currently being proceeded by the International Methane Emissions Observatory (IMEO). In the framework of the project, such instruments are to be installed during a field campaign planned for the fall of 2022 in selected mines of USCB.

* Open path instruments measure the averaged methane concentration at the shaft cross-section. In contrast, a closed path analyser measures the methane concentration at a single point at the exhaust of the ventilation shaft. Please note that the second type should be sampling at different locations inside the shaft in order to provide information on the homogeneity (or lack thereof) of the air stream.

**Reviewer 2, Specific Comments**

**Issue 3**: Line 29 – what was the uncertainty in the number 142.68 kt/yr?

*The standard deviation for the number 142.68 kt yr$^{-1}$ is σ =18.63 kt yr$^{-1}$. It was added in line in L28.*

**Issue 4:** Line 83 – not clear what you mean by 'both' – do you mean both top down and bottom-up

*The sentence was rewritten—line L86.*

**Issue 5:** Line 110 – do you mean individual rather than particular?

*We meant individual. It was corrected.*

**Issue 6:** Line 138 – for which year?

*The data are for 2018. It was added to the sentence in line L160*

**Issue 7:** Line 144 – what is meant by the levels of methane concentrations – is that atmospheric methane concentrations, or concentrations within the mines?

*It was not about the concentration level but about the methane content, i.e. the actual methane content (in m$^3$), which defines the volume of natural methane included in one tonne of dry ash-free coal without ash and moisture content (tonne daf). The sentence was corrected – L171-172*

**Issue 8:** Figure 1 isn't a particularly useful figure, these numbers could just be given in the text.

*The authors removed Figure 1 and added the description in the text - see L153-158.*

**Issue 9:** Figure 2 needs a more detailed caption to explain it. How was methane emission found in the previous studies. Could you add a scale to the map?

*The drawing is for illustration only. Its purpose was to illustrate how the exploitation fields of individual mines are located on the territory of the USCB. It is not made to scale. The authors enlarged the font in the drawing to make the names more visible.*

*The authors added information about previous research in the text – L165-172.*

**Issue 10:** Line 193 – what is meant by 'the last parameter'?

*We meant methane content. The sentence was rewritten – L213*

**Issue 11:** Line 306 – I think you give the dates of the CoMET mission 4 times in the paper – it doesn't need repeating.

*Corrected.*

**Issue 12:** Line 320 – what is the reference for these emissions?

*The authors added the reference – L340*

**Issue 12:** Line 342 – do you mean high frequency concentration measurements would be a helpful tool to measure emissions?

*Yes, of course. It was corrected – L365*

**Issue 13:** Line 371 – what is the high and low concentration range?

*Authors added the explanation – L392-393 and 410-412*

**Issue 14:** Line 390 – what is meant by 'joint exhaust'?

*The authors meant:* the collective airflows of the return air. It was corrected – L416-417

**Issue 15:** Figure 6 – it would be useful to annotate this.

*The authors added the description to figure – L446-449*

Equation 2 – how can it go from $m^3$/min on the left hand side to $m^3$/s on the right hand side (need to add a conversion factor)?

*The equation was corrected. The conversion factor was added – L460*

**Issue 16:** Is the air flow velocity sensor in the middle of the flow. Does is make any difference if it's positioned near the edges or not?

*The speed probe is located near the edge of the ventilation channel. Considering the very high flow velocities, which can reach 23,000 $m^3$ / min, it is impossible to install it on the axis of the channel, as it would be damaged. It should be noted that with such large air velocities shaft, there is a turbulent flow, so the lateral location of the device should not affect its indications.*

*In order to make sure that the flow reading is correct, mine operators should always compare the readings of the air flows in the channel of the ventilation shaft with the measurements taken at the locations of the concentration measurements, i.e. at the intersections of the collective airflows of return air flowing into the shaft at its bottom – the sum of the latter should be equal to the former, according*

*to mass balance. Unfortunately, we were not able to obtain the results of these comparisons and were informed that they are not recorded.*

**Issue 16:** Figure 9 – most of the measurements have 0.1% precision, but there are some measurements that appear to have a smaller precision. Why is this? Is there any information available about operations at the mine that would account for some of the variability seen?

*In the case of the Pniówek mine (to which the graph refers), observed variation in methane concentrations resulted mainly from the scope of mining works. At that time, the mine had high methane prone longwalls. Excavation of these resulted in numerous technological breaks caused mainly to maintain safety of the mining crews.*

*As the concentration fluctuations is correlated to the mining activity, methane outflows from the goaves could have occurred, as evidenced by increased methane concentrations. In addition, pressure changes could have resulted in an increased emission of methane, which was immediately visible on the sensors, alerting the personnel responsible for crew safety. During technological breaks, methane emissions were almost ceased.*

**Issue 17:** In table 4 it's not clear why 2 of the sites have 2 different values in the temporal data column. i.e. it looks like there are 2 measurements for Zofiówka and Knurów. I think you've grouped together some of the shafts and added their emissions in the second column but that needs to be clearer.

*The table has been reformatted and description clarified.*

*For further explanation of the administrative groupings of mines, please see response to the question from Reviewer 1, Issue #7.*

**Issue 18:** Line 631 – according to table 4 some of the mines had emissions lower than 60.02 kg/min. Where did this number come from? I think these are the 2 highest emissions of the mines, not the range in emissions.

*We assume that the reviewer meant Table 3, where statistics of emissions for individual shafts are presented.*

*The values provided in L631 are not provided in the table directly. Rather, they should correspond to the sum of average amounts of methane discharged by these two mines over all their respective shafts over the entire period considered. A minor mistake in calculation caused the sum from the table to not be equal to the numbers provided. The authors apologize for this mistake.*

*In the revised manuscript, the authors removed these numbers for clarity and now it is only stated that these two mines discharged the highest amount of methane in the studied period – L618-619.*

---

## Referee Report (RR1)

The paper is much improved since the earlier discussion version.

It gives a clear explanation and demonstration of a technique to monitor coal mine methane emissions, which would be valuable to publish.

I just noticed 3 minor technical errors to be corrected before publication.

Figure 2 – typographical error in the y axis label methane content

Line 1129. Superscript the 3 and -1 to format the units correctly.

Line 2486 (the last sentence) doesn't make sense – missing 'be' in these can be intermittent rather than continuous?

---

## Author Response (AR2)

The authors would like to thank the reviewers for the positive review of the article. All suggested corrections have been included by the authors.